# Improved figure of merit ($z$) at low temperatures for superior thermoelectric cooling in Mg$_3$(Bi,Sb)$_2$

Nan Chen[1,2], Hangtian Zhu [1] ✉, Guodong Li [1], Zhen Fan[1], Xiaofan Zhang[1,2], Jiawei Yang[1,2], Tianbo Lu[1], Qiulin Liu[1,2], Xiaowei Wu[1,2], Yuan Yao [1], Youguo Shi[1] & Huaizhou Zhao [1] ✉

The low-temperature thermoelectric performance of Bi-rich n-type Mg$_3$(Bi,Sb)$_2$ was limited by the electron transport scattering at grain boundaries, while removing grain boundaries and bulk crystal growth of Mg-based *Zintl* phases are challenging due to the volatilities of elemental reactants and their severe corrosions to crucibles at elevated temperatures. Herein, for the first time, we reported a facile growth of coarse-grained Mg$_3$Bi$_{2-x}$Sb$_x$ crystals with an average grain size of ~800 μm, leading to a high carrier mobility of 210 cm$^2 \cdot$ V$^{-1} \cdot$ s$^{-1}$ and a high $z$ of $2.9 \times 10^{-3}$ K$^{-1}$ at 300 K. A $\Delta T$ of 68 K at $T_h$ of 300 K, and a power generation efficiency of 5.8% below 450 K have been demonstrated for Mg$_3$Bi$_{1.5}$Sb$_{0.5}$- and Mg$_3$Bi$_{1.25}$Sb$_{0.75}$-based thermoelectric modules, respectively, which represent the cutting-edge advances in the near-room temperature thermoelectrics. In addition, the developed grain growth approach can be potentially extended to broad *Zintl* phases and other Mg-based alloys and compounds.

Thermoelectrics (TEs) can directly convert heat into electricity, or vice versa, and enable broad applications in solid-state cooling and power generation[1–3]. Generally, the performance of thermoelectric device is governed by the materials' figure of merit $z$, which can be expressed as $z = S^2\sigma/(\kappa_L + \kappa_e)$, where $S$, $\sigma$, $\kappa_L$, and $\kappa_e$ are the Seebeck coefficient, electrical conductivity, lattice thermal conductivity, and electronic thermal conductivity, respectively[4]. Here, the Seebeck coefficient ($S$) and electrical conductivity ($\sigma$) are coupled through the carrier concentration ($n$) and the band effective mass ($m^*$)[3]. Among the strategies to decouple $S$ and $\sigma$ for enhanced power factor ($PF = S^2\sigma$), various strategies have been employed, including the resonant level doping[5] and band convergence[6]. However, for material systems with intrinsically low lattice thermal conductivity, such as SnSe(S)[7], Cu$_2$Se(S,Te)[8], MgAgSb[9], etc., the grain growth has been proposed as an effective strategy to enhance the carrier mobility and figure of merit $z$, through reduced grain-boundary scattering and having nearly no impacts on Seebeck coefficient.

The high thermoelectric performance of n-type Mg$_3$(Bi,Sb)$_2$ originated from its large degeneracy number of the conduction band up to six[10,11] and the intrinsically low lattice thermal conductivity of 0.497 W $\cdot$ m$^{-1} \cdot$ K$^{-1}$ for Mg$_3$Sb$_2$ and 0.349 W $\cdot$ m$^{-1} \cdot$ K$^{-1}$ for Mg$_3$Bi$_2$[10]. Among these, Bi-rich alloys possess relatively higher $z$ values around room temperatures (150–550 K), given the reason that the small effective mass of narrow bandgap material favors high carrier mobility at low temperatures[12,13]. It is revealed that the carrier scattering mechanism of n-type Mg$_3$(Bi,Sb)$_2$ was dominated by grain boundaries[14–18] and several methods on single crystal growth have been reported[16,19–21]. However, unfortunately, the above methods involve complicated apparatus[19] and multiple experimental steps[16,21], and the impurities introduced by high-temperature processes can severely degrade TE properties. Moreover, the irregular shapes and limited single crystal sizes (lamellar shape with a thickness of 0.2–1.2 mm) add extra difficulties in thermoelectric device fabrications. Therefore, the facile growth of high-quality Mg$_3$(Bi,Sb)$_2$ crystals remains to be challenging.

[1]Beijing National Laboratory for Condensed Matter Physics, Institute of Physics, Chinese Academy of Sciences, Beijing 100190, China. [2]College of Materials Science and Opto-Electronic Technology, University of Chinese Academy of Sciences, Beijing 100049, China. ✉e-mail: htzhu@iphy.ac.cn; hzhao@iphy.ac.cn

Grain growth usually performs at high temperatures, posing difficulties in deliberate control of the content of volatile reactants and meanwhile the raw materials could react with the crucible, leading to deviation from the stoichiometry of the target composition. To solve this problem, we hereby developed a universal approach to synthesize a variety of coarse-grained alloys or compounds containing active elements, *e.g.* Mg and Sb, represented by the *Zintl* phase $Mg_3(Bi,Sb)_2$. Following this method, we have prepared bulk polycrystal n-type Bi-rich $Mg_3Bi_{2-x}Sb_x$ ($x$ = 0.5, 0.75) with grain size up to 1.0 mm. Such grain size encompasses almost two-third of the length of a thermoelectric leg in the devices. The chemical composition, defects and grain size of $Mg_3Bi_{2-x}Sb_x$ materials were successfully regulated to approach the limit of carrier mobility for a single crystal. Combined with the intrinsically low lattice thermal conductivity[22,23], the $z$ values of both $Mg_3Bi_{1.5}Sb_{0.5}$ and $Mg_3Bi_{1.25}Sb_{0.75}$ increased significantly below and at room temperatures, which is beneficial for the cooling purpose. Moreover, the as-grown coarse-grained bulk crystals (with a dimension of $\varnothing$12.7mm × 13mm) also exhibit excellent uniformity and mechanical performance, bringing great advantages for subsequent thermoelectric device fabrication in this work. This technique is also applicable for the crystal growth of MgAgSb[9], *Zintl* phase[24], and other Mg-based alloys[25], which contain reactive and volatile elements (such as alkaline metals, and rare earth elements).

## Results and discussion

### Growth of coarse-grained bulk crystal and microstructure characterizations

To better control the synthesis of $Mg_3(Bi,Sb)_2$ crystals, a $ZrO_2$/Ta/$SiO_2$ three-layer nested-crucible (Supplementary Fig. 1) was developed to seal and load the highly reactive and corrosive elements, Mg and Sb (details available in Methods). A temperature procedure for the synthesis and growth of bulk $Mg_3Bi_{2-x}Sb_x$ ($x$ = 0.5, 0.75) was designed based on the pseudo-binary phase diagram of Mg-$Bi_{0.75}Sb_{0.25}$[26,27], as shown in Fig. 1a. The crystal growth process involved the following steps: The samples were placed in a rocking furnace and initially heated to 1273 K to melt the precursors. The molten sample was continuously swung for 2 h to ensure a complete reaction and the formation of a uniform liquid phase. Subsequently, the samples were slowly cooled down to 1193 K at a rate of 2 K/h and held at this temperature for 24 h. At 1193 K, the high-temperature $Mg_3(Bi,Sb)_2$ phase began to solidify from the melt state. Nucleation is likely initiated at specific sites in the crucible, such as the junction of the sidewall and the bottom of the crucible. The 24-hour holding period facilitated the preservation of high-quality nuclei for subsequent grain growth, while low-quality nuclei gradually shrank and melted away. Next, the crucible was further cooled down to 1073 K at a rate of 2 K/h and annealed for 24 h. At this stage, the amount of the high-temperature phase (cubic phase) gradually increased by consuming the liquid phase during cooling. The small amount of remaining liquid phase between the grains promoted grain growth during the annealing process, as depicted in the inset of Fig. 1a. Following this, the crucible was cooled down to 973 K at a rate of 2 K/h and held at this temperature for 72 h. Throughout this step, the high-temperature phase completely transformed into the uniform hexagonal phase (room temperature phase) of $Mg_3Bi_{2-x}Sb_x$ ($x$ = 0.5, 0.75). The longtime annealing helped eliminate defects in the material. Finally, the crucible was allowed to naturally cool down to room temperature. The obtained coarse-grained $Mg_{3.15}Bi_{1.4975}Sb_{0.5}Te_{0.0025}$ and $Mg_{3.15}Bi_{1.2475}Sb_{0.75}Te_{0.0025}$ samples are referred to as CGBi1.5 and CGBi1.25 in this context, respectively. The measured densities of as-grown CGBi1.5 and CGBi1.25 materials are 5.36 g/cm³ and 5.14 g/cm³, respectively, agreeing well with the experimental and theoretical values[17,19] (Supplementary Table 1).

EBSD was employed to characterize the grain morphology of the as-grown CGBi1.5 bulk material. Contrast with the small grain sizes of

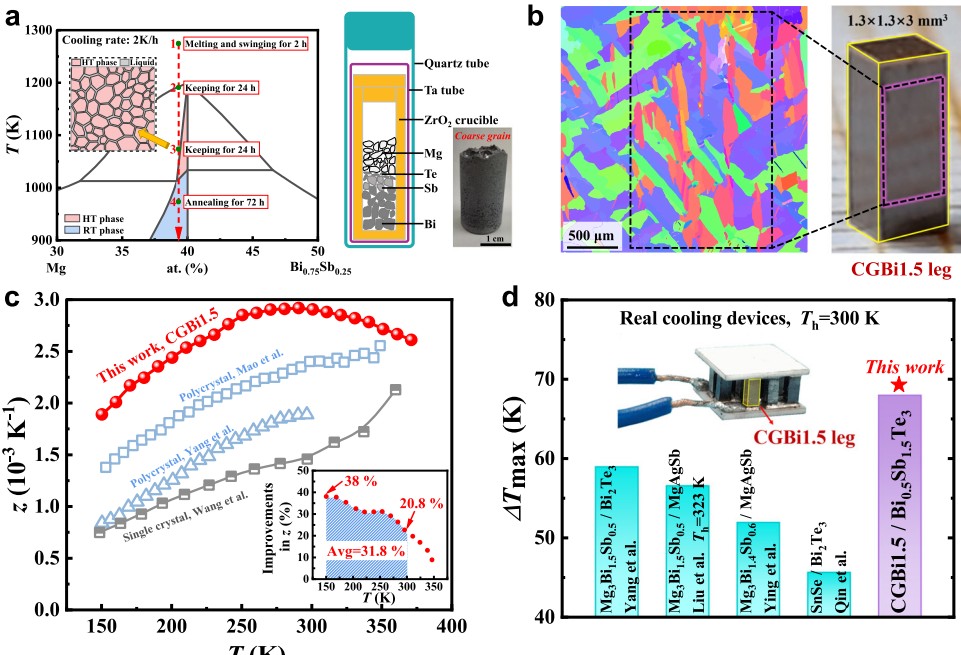

**Fig. 1 | The synthesis and growth mechanism of coarse-grained $Mg_3Bi_{2-x}Sb_x$ ($x$ = 0.5, 0.75) bulk material and characterization of the thermoelectric performances for CGBi1.5 and the relevant TE modules. a** Schematic of the pseudo-binary $Mg$-$Bi_{0.75}Sb_{0.25}$ phase diagram, the synthesis procedures, and the setup of the crucibles. The picture of as-grown coarse-grained $Mg_3Bi_{2-x}Sb_x$ bulk crystals was shown on the right. **b** The electron back-scattering diffraction (EBSD) image of as-grown coarse-grained crystals, and the photograph of a single CGBi1.5 leg. **c** The measured $z$ of as-grown CGBi1.5 sample as a function of temperature, compared to literatures[19,30,31]. The inset shows the percentage of improvements in $z$ compared with the data reported by Mao et al.[31] **d** Comparison of the maximum $\Delta T$ of real cooling devices, including our CGBi1.5/$Bi_{0.5}Sb_{1.5}Te_3$ cooling module and new material-based modules[30,35–37]. Inset shows the as-fabricated modules.

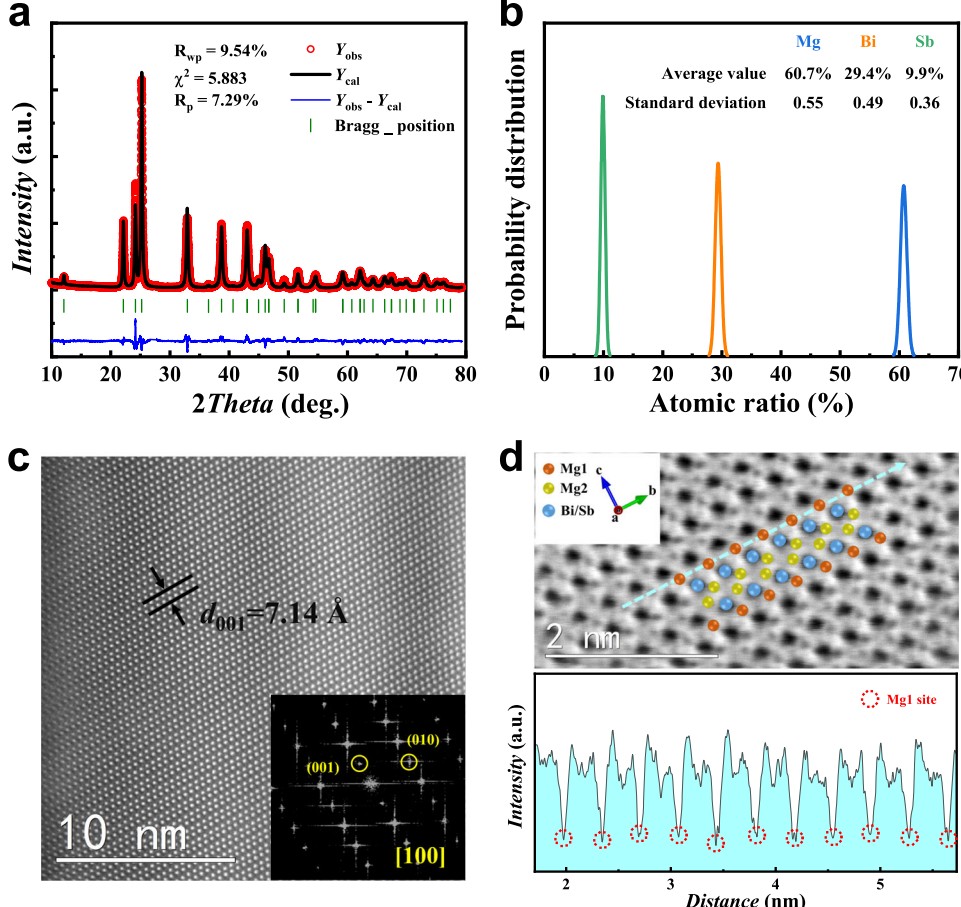

**Fig. 2 | Crystal and microstructure characterizations of CGBi1.5. a** Rietveld refinement against the X-ray diffraction patterns at 300 K for CGBi1.5. **b** The probability distribution of elemental compositions at different regions of a 12.7 mm diameter CGBi1.5 pellet. **c** The high-resolution scanning transmission electron microscopy (STEM) image of CGBi1.5, with the fast Fourier transform (FFT) diffractogram shown in the inset. **d** The STEM image of CGBi1.5 and line profile along the dashed blue line marked in the image.

~20 μm observed in the spark plasma sintered (SPS) sample (Supplementary Fig. 2), the obtained bulk ingot has an average grain size of ~800 μm, and with some regions reaching ~1.0 mm as shown in Fig. 1b. Here the large grain sizes in CGBi1.5 bulk material exhibited a typical feature of the coarse grain[28]. Considering the dimensions (~1.3 × 1.3 × 3 mm³) of thermoelectric legs in our cooling device, ideally the transport of electrons in one single leg can be scattered by only two grain boundaries, which is analogy to the scenarios in a single crystal. Moreover, the as-grown CGBi1.5 bulk materials show a Vickers hardness of 0.54 GPa, close to that of 0.61 GPa for $Mg_{3.065}Sb_{1.3}Bi_{0.7}Gd_{0.015}$ as reported in the literature[29]. The compressive strength of the CGBi1.5 sample is greater than 300 MPa, because no cracks occur under the loading stress of 300 MPa, which is consistent with the literature results[29], as shown in Supplementary Fig. 3. The above results affirm the excellent mechanical properties of our coarse-grained materials, fulfilling the needs for TE device manufacturing.

The obtained bulk material was cut and polished for the measurement of thermoelectric properties. Figure 1c and Supplementary Fig. 4 show that the CGBi1.5 sample exhibited a peak $z$ of $2.92 \times 10^{-3}$ K$^{-1}$ at 290 K, which was the highest among literatures for $Mg_3Bi_{1.5}Sb_{0.5}$ materials to the best of our knowledge[19,30,31]. Most of the thermoelectric cooling devices work in the temperature range of 150–300 K, where the coarse-grained CGBi1.5 demonstrated remarkable advantage over other $Mg_3(Bi,Sb)_2$ based materials, e.g., the average $z$ is 31.8% higher than the data reported by Mao et al.[31], indicating a great potential for thermoelectric cooling applications[32–34], as shown in the

inset of Fig. 1c. A 7-pair thermoelectric cooling module was fabricated based on above CGBi1.5 and the commercial p-type $Bi_{0.5}Sb_{1.5}Te_3$. A cooling temperature difference (Δ$T$) of 68 K was obtained at the hot-side temperature ($T_h$) of 300 K, which is the largest among the recently reported full-scale new material-based real cooling devices such as $Mg_3(Bi,Sb)_2/Bi_2Te_3$[30], $Mg_3(Bi,Sb)_2/MgAgSb$[35,36], and $SnSe/Bi_2Te_3$[37] (Fig. 1d). It is noted that the temperature difference was underestimated by ~2 K due to the use of alumina ceramic plates.

The Rietveld refinement against experimental powder X-ray diffraction (XRD) data identified the formation of phase-pure hexagonal $Mg_{3.15}Bi_{1.4975}Sb_{0.5}Te_{0.0025}$ (space group $P\bar{3}m1$, No. 164, $a = 4.6652(1)$ Å, $c = 7.4058(2)$ Å; Fig. 2a). The Rietveld refinement also suggested a site occupancy factor (SOF) of Mg1 of 92 ± 1%, i.e., corresponding to the stoichiometry of $Mg_{2.92(1)}Bi_{1.4975}Sb_{0.5}Te_{0.0025}$, which is in the similar range and slightly higher than that of 89 ± 2% as reported for $Mg_{3.2}Sb_{1.5}Bi_{0.49}Te_{0.01}$[52], revealing the possible enhanced n-type carrier transportation in our sample. The phase purity of the as-grown CGBi1.25 sample was also confirmed by the XRD as shown in Supplementary Fig. 5. To confirm the composition uniformity of the material, EDS analyses were performed for a total of nine regions from a 12.7 diameter CGBi1.5 pellet. Figure 2b shows the probability distribution of the atomic ratios across these regions, which confirms the uniform elemental distributions across the whole area and is consistent with the elemental mapping results as shown in Supplementary Fig. 6. The high-resolution scanning transmission electron microscopy (STEM) image in Fig. 2c showed the regular arrangement of atoms in the

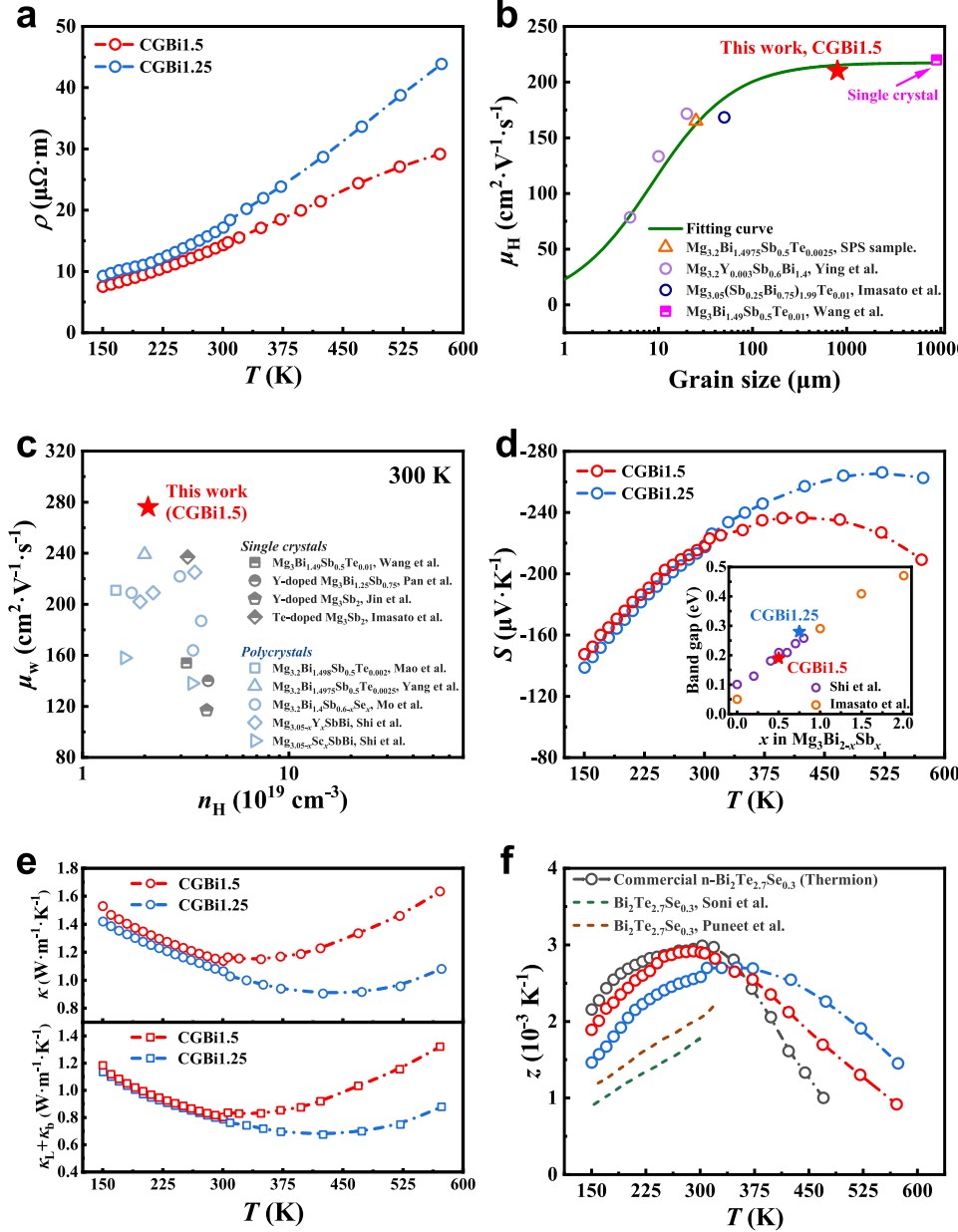

**Fig. 3 | Thermoelectric properties of CGBi1.5 and CGBi1.25. a** Temperature-dependent electrical resistivity. **b** Carrier mobility of n-type $Mg_3(Bi,Sb)_2$ as a function of grain size. **c** Weighted mobility in comparison to literature values[16, 19–21, 30, 31, 53–55]. **d** Temperature-dependent Seebeck coefficient with the calculated bandgap in comparison to the literatures[12, 13] shown in the inset. **e** Temperature-dependent thermal conductivity, lattice and bipolar thermal conductivity. **f** z values compared with state-of-the-art n-type $Bi_2Te_{2.7}Se_{0.3}$ materials[40, 41].

selected area, indicating a high crystal quality, and the inset fast Fourier transform (FFT) image is well indexed to [100] zone axis. Both heavy Bi/Sb atoms and light Mg atoms were observed in Fig. 2d, and the line profile along the dashed blue line indicated consistent and stable Mg1 intensities, which reinforced the high Mg1 occupancy.

## Thermoelectric property characterization

As shown in Fig. 3a, the increasing of electrical resistivity with temperature shows metallic behavior of CGBi1.5 and CGBi1.25 samples with a respective carrier concentration of $2.08 \times 10^{19} \, cm^{-3}$ and $2.2 \times 10^{19} \, cm^{-3}$. Owing to the reduced grain-boundary scattering, the electrical resistivity of CGBi1.5 samples at room temperature was measured to be $14.3 \, \mu\Omega \, m$, decreased by ~25% compared to that of SPS-processed polycrystalline $Mg_{3.2}Bi_{1.4975}Sb_{0.5}Te_{0.0025}$ samples ($19.2 \, \mu\Omega \cdot m$) with a similar carrier concentration[30]. The literature data

of grain sizes ($d$) and corresponding carrier mobilities ($\mu_H$) of Bi-rich n-type $Mg_3(Bi,Sb)_2$ materials[12,19,36] were collected and compared with the measured data in this work, as shown in Fig. 3b. A grain-boundary dominated transportation model was adopted to unveil the $\mu_H \sim d$ relation (details available in Methods). The experiment data in this work was in good agreement with the fitting curve, and the height of the potential barrier of 170 meV was obtained for the grain-boundary scattering. As the grain size increases, the effect of grain-boundary on the mobility decreases rapidly, and the CGBi1.5 coarse grain with an average grain size of ~800 μm exhibits a high carrier mobility ($210 \, cm^2 \cdot V^{-1} \cdot s^{-1}$) approaching the limit for a single crystal[19], indicating the high quality of the crystallization of the coarse-grained bulk materials.

Given the difference in density of states' effective mass ($m_d^*$) with varied Bi/Sb ratio, the weighted mobility ($\mu_w$) can better describe the

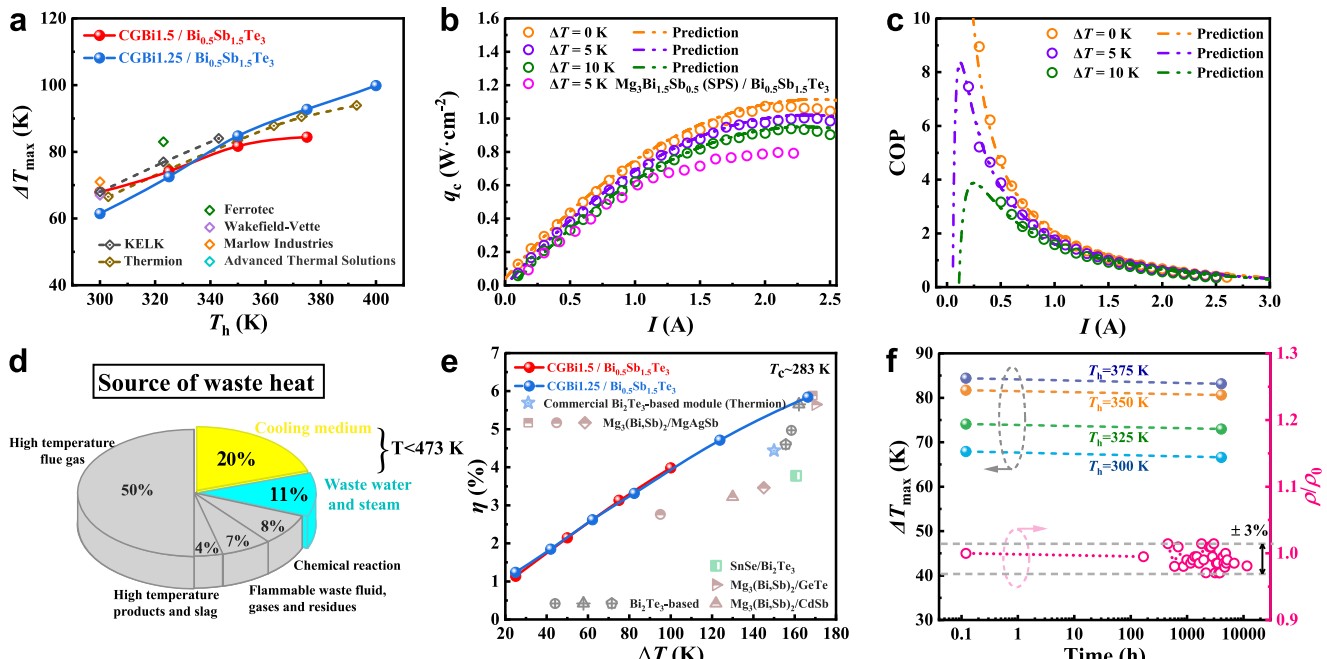

**Fig. 4 | Cooling and power-generation performances of $Mg_3Bi_{2-x}Sb_x$/ $Bi_{0.5}Sb_{1.5}Te_3$ ($x = 0.5, 0.75$) based TE modules. a** The comparison of maximum cooling temperature differences between $Mg_3Bi_{2-x}Sb_x$/$Bi_{0.5}Sb_{1.5}Te_3$ ($x = 0.5, 0.75$) TE modules and commercial $Bi_2Te_3$-based module with a variety of hot-side temperatures $T_h$. **b** The cooling power density ($q_c$) as a function of current for CGBi1.5/ $Bi_{0.5}Sb_{1.5}Te_3$ based modules, in comparison to the 7-pair SPS $Mg_3Bi_{1.5}Sb_{0.5}$/ $Bi_{0.5}Sb_{1.5}Te_3$ cooling module with the normalized cross-section. **c** COP for CGBi1.5/ $Bi_{0.5}Sb_{1.5}Te_3$-based modules at temperature differences of 0 K, 5 K, and 10 K, respectively. All the dashed dot lines represent the theoretical prediction and the open circles indicate experimental data in **b**, **c**. **d** The distribution of industrial waste heat, where the waste heat at $T < 473$ K (mostly cooling medium and waste water and steam) takes up ~31%[56]. **e** The measured conversion efficiencies as a function of temperature difference for the $Mg_3Bi_{2-x}Sb_x$/$Bi_{0.5}Sb_{1.5}Te_3$ ($x = 0.5, 0.75$) TE modules compared to that of commercial $Bi_2Te_3$ and results from literatures based on $Bi_2Te_3$[42–44], SnSe[37] and $Mg_3(Bi,Sb)_2$ materials[35, 36, 45–47]. **f** Aging time dependent maximum cooling temperature differences of CGBi1.5/$Bi_{0.5}Sb_{1.5}Te_3$ devices and electrical resistivity of CGBi1.5 samples.

inherent carrier transport properties[38]. As shown in Fig. 3c, the weighted mobility of coarse-grained CGBi1.5 crystals (276 cm$^2 \cdot$V$^{-1} \cdot$s$^{-1}$) was the highest among the n-type $Mg_3(Bi,Sb)_2$ materials. Despite the similar $\mu_H$, the $\mu_w$ of CGBi1.5 sample was higher than that of the single crystal with the same Bi/Sb ratio. Since the grain size nearly has no impact on the effective mass of conduction band, the Seebeck coefficient of coarse-grained CGBi1.5 crystals was very close to that of polycrystals with the same carrier concentration[30]. As shown in the inset of Fig. 3d, the Goldsmid-Sharp bandgap $E_g$ was calculated from the maximum Seebeck coefficient and the temperature at which it occurs[39]. The Seebeck coefficient of CGBi1.5 and CGBi1.25 reached their peak values at 400 K and 525 K, and the bandgaps are 0.19 eV and 0.28 eV, respectively, which was consistent with polycrystalline materials with the same Bi/Sb ratio[12,13], further confirming the excellent uniformity and high quality of coarse-grained crystals. Owing to a wider bandgap, the Seebeck coefficient of CGBi1.25 was less affected at elevated temperature. The enhanced electrical conductivity together with the high Seebeck coefficient enabled the high power factor of 33 $\mu W \cdot$cm$^{-1} \cdot$K$^{-2}$ for CGBi1.5, and 27.5 $\mu W \cdot$cm$^{-1} \cdot$K$^{-2}$ for CGBi1.25 at 300 K, respectively, as shown in Supplementary Fig. 7.

Figure 3e shows the combination of lattice and bipolar thermal conductivity ($\kappa_L + \kappa_b$) as the function of temperatures, and the value of 0.80 W$\cdot$m$^{-1} \cdot$K$^{-1}$ for CGBi1.5 at 300 K is comparable to that of reported values, i.e., 0.796 W$\cdot$m$^{-1} \cdot$K$^{-1}$ for $Mg_3Bi_{1.49}Sb_{0.5}Te_{0.01}$ single crystal[19] and 0.79 W$\cdot$m$^{-1} \cdot$K$^{-1}$ for $Mg_{3.2}Bi_{1.498}Sb_{0.5}Te_{0.002}$ polycrystal[31]. The results indicated that the grain-boundary scattering can pose significant impact on the carrier transportation through the potential barrier located at the grain-boundary, while having less effect on the phonons. This mainly because the phonon is less coupled with the electric field and its mean free path is much smaller than the grain size.

The improved electrical transport and intrinsically low lattice thermal conductivity significantly improved the $z$ to $2.9 \times 10^{-3}$K$^{-1}$ for CGBi1.5 at 300 K, as can be seen from Fig. 3f, which is comparable to state-of-the-art commercial n-type $Bi_2Te_{2.7}Se_{0.3}$ materials ($z \sim 2.94 \times 10^{-3}$K$^{-1}$ at 300 K) and outperformed most of the values reported in the literatures[40,41]. Thermoelectric properties of commercial $Bi_2Te_3$ materials were shown in Supplementary Fig. 8. As for CGBi1.25, the peak $z$ reaches ~$2.7 \times 10^{-3}$K$^{-1}$ at the temperature of 350 K. The electrical transport properties of CGBi1.5 samples in different parts and from different samples have been tested (Supplementary Fig. 9 and 10), attesting to the uniform thermoelectric performance achieved through our method. Supplementary Fig. 11 reveals the good reproducibility of this synthesis method represented by the thermoelectric properties of several batches of CGBi1.5 samples. The $z$ of CGBi1.5 in 150–300 K is comparable to that of commercial n-type $Bi_2Te_{2.7}Se_{0.3}$, indicating its potentially high cooling performance.

## Performance of thermoelectric modules

For CGBi1.5/$Bi_{0.5}Sb_{1.5}Te_3$-based module, the measured $\Delta T_{max}$ reached 68 K, 74 K, and 82 K with the hot-side temperatures of 300 K, 325 K, and 350 K, respectively, as shown in Fig. 4a and Supplementary Fig. 12, which are the highest among all the cooling devices with measurable cooling power density based on $Mg_3Bi_{1.5}Sb_{0.5}$[30,35], $Mg_3Bi_{1.4}Sb_{0.6}$[36] and SnSe[37] materials (Supplementary Fig. 13), and also comparable to the state-of-the-art commercial $Bi_2Te_3$ cooling module. The difference in $\Delta T_{max}$ curve between the modules reported in this work and commercial $Bi_2Te_3$-based devices can be attributed to the higher near-room temperature thermoelectric properties of $Bi_2Te_3$ and the advanced contact layer fabrication technique in the leading thermoelectric enterprise. It is worth noting that Mao et al. reported a similar $\Delta T$ of 69 K at $T_h = 300$ K with a unicouple TE cooling device based on

$Mg_{3.2}Bi_{1.498}Sb_{0.5}Te_{0.002}$ and $Bi_{0.5}Sb_{1.5}Te_3$, while the relevant cooling capacity ($Q_c$) and coefficient of performance (COP) were hard to obtain because of the large uncertainty in the measurement of small heat flow. As for CGBi1.25/$Bi_{0.5}Sb_{1.5}Te_3$ module, despite the relatively low $\Delta T$ of 62 K at $T_h$ = 300 K, the higher $z$ values of CGBi1.25 at 350–400 K lead to larger $\Delta T$ at high temperatures, i.e., 85 K at $T_h$ of 350 K, and 100 K at $T_h$ of 400 K, as shown in Fig. 4a and S12. Therefore, these two kinds of $Mg_3Bi_{2-x}Sb_x$/$Bi_{0.5}Sb_{1.5}Te_3$ ($x$ = 0.5, 0.75) TE modules could substitute commercial $Bi_2Te_3$ modules over a wide temperature range.

As shown in Fig. 4b, the thermoelectric module created a larger $q_c$ at a smaller $\Delta T$ between the two sides of the module because less cooling heat was dissipated through thermal conduct in the materials. The heat flow increased gradually with the working current and reached its maximum cooling power density ($q_{c,max}$). With the increase of temperature difference from 0 K to 10 K, the $q_{c,max}$ decreased from 1.07 W cm$^{-2}$ to 0.94 W cm$^{-2}$ at the optimal current of around 2.3 A. In this work, the normalized $q_{c,max}$ of 1.0 W cm$^{-2}$ at $\Delta T$ = 5 K is 25% higher than that of the cooling module made of SPS-processed $Mg_3Bi_{1.5}Sb_{0.5}$ and commercial $Bi_{0.5}Sb_{1.5}Te_3$ materials, which is originated from the higher power factor of CGBi1.5 samples (Supplementary Fig. 7). COP, another important parameter of cooling module, defined as the ratio of $Q_c$ to the input power $p$, is critical to evaluate the energy conversion efficiency. The measured maximum COP of CGBi1.5/$Bi_{0.5}Sb_{1.5}Te_3$ module reached 8.9, 7.5, and 3.2, at the temperature difference of 0 K, 5 K, and 10 K, respectively, as shown in Fig. 4c. These values were higher than the $Mg_3(Bi,Sb)_2$/MgAgSb module[35] (COP$_{max}$ = 2.6 at $\Delta T$ of 10 K) and comparable to the state-of-the-art commercial $Bi_2Te_3$ module (COP$_{max}$ = 8.8 at $\Delta T$ of 0 K) (as shown in Supplementary Fig. 14).

It is known that industrial waste heat can be classified into six categories based on the sources, as illustrated in Fig. 4d. Among them, waste heat below 473 K occupies ~31% of total waste heat. As for the power generations of CGBi1.25/$Bi_{0.5}Sb_{1.5}Te_3$ modules, a conversion efficiency as high as 5.8% was demonstrated within the temperature range of 280–450 K, as shown in Fig. 4e. At the $\Delta T$ of 100 K, both $Mg_3Bi_{2-x}Sb_x$/$Bi_{0.5}Sb_{1.5}Te_3$ ($x$ = 0.5, 0.75) modules exhibit similarly high conversion efficiencies, which is originated from their comparable average $z$ values in the temperature range of 280–390 K, as indicated in Fig. 4e and Supplementary Fig. 4. The efficiencies at a variety of $\Delta T$ from 20 to 170 K was the highest among those of reported modules based on $Bi_2Te_3$ system[42–44], $Mg_3(Bi,Sb)_2$/MgAgSb[35,36,45], SnSe/$Bi_2Te_3$[37] and so on[46,47].

Additionally, a $\Delta T$ of 25 K in this module could produce an output power of 4 mW with a conversion efficiency >1%, which can use the tiny temperature difference between human skin and the environment to support the operation of wearable electronic devices[48]. With the waste heat source of 450 K, an output power density up to 0.14 W·cm$^{-2}$ can be produced by this high-efficiency CGBi1.25/$Bi_{0.5}Sb_{1.5}Te_3$ based module ($\eta$~5.8%). The open-circuit voltage ($V_{oc}$) and internal resistance ($R_{in}$) as a function of $T_h$ of $Mg_3Bi_{2-x}Sb_x$/$Bi_{0.5}Sb_{1.5}Te_3$ ($x$ = 0.5, 0.75) modules are shown in Supplementary Fig. 15 and 16. The measured $V_{oc}$ is consistent well with the prediction from the TE properties. The discrepancy between the experimental and predicted values of $R_{in}$ was ascribed to the contact resistance (13.5 µΩ·cm$^2$ for $Mg_2Cu$/CGBi1.5 interfaces, as shown in Supplementary Fig. 17). Therefore, high thermoelectric performance of the as-grown bulk Bi-rich $Mg_3Bi_{2-x}Sb_x$ ($x$ = 0.5, 0.75) materials was confirmed, revealing great potentials for both cooling and power generation at near-room temperatures.

As shown in Fig. 4f, the maximum temperature differences of CGBi1.5/$Bi_{0.5}Sb_{1.5}Te_3$ devices were only reduced by *ca.* 1 K after being stored in the glove box for nearly 6 months. In addition, we also observed that with reduced Mg1 vacancies, the electrical resistivity of coarse-grained CGBi1.5 bulk materials stored in the glove box

remained unchanged over 16 months, which, to the best of our knowledge, has been the best record to date. These reinforce the excellent stabilities of both the thermoelectric materials and electrodes of $Mg_3Bi_{2-x}Sb_x$/$Bi_{0.5}Sb_{1.5}Te_3$ ($x$ = 0.5, 0.75) modules. The results also provide feasibility to further improve device-level stabilities through subsequent device packaging strategies, underpinning potential real-world applications.

In conclusion, two coarse-grained $Mg_3Bi_{2-x}Sb_x$ ($x$ = 0.5, 0.75) bulk crystals were synthesized and grown through a high-temperature melting approach. The as-grown CGBi1.5 crystals exhibit a high carrier mobility and low lattice thermal conductivity, leading to a high $z$ of -2.9 × 10$^{-3}$K$^{-1}$ at room temperature. Meanwhile, this material also exhibits the highest average $z$ value of -2.57 × 10$^{-3}$K$^{-1}$ in the temperature range of 150–300 K among n-type $Mg_3(Bi,Sb)_2$ materials with the same Bi/Sb ratio. The 7-pair cooling module made of n-type CGBi1.5 and p-type commercial $Bi_{0.5}Sb_{1.5}Te_3$ yielded a high $\Delta T$ of 68 K with the hot-side temperature of 300 K. Besides that, the CGBi1.25/$Bi_{0.5}Sb_{1.5}Te_3$ devices displayed a high power-generation efficiency of 5.8% within the temperature range of 280–450 K. This work realized the growth of coarse-grain $Mg_3Bi_{2-x}Sb_x$ bulk crystals through the nested-crucible, and facilitate its applications in high-performance thermoelectric cooling and power-generation modules. Moreover, the grain growth approach for enhancing thermoelectric properties, as developed in this study, shows promise for broader applications in *Zintl* phases, along with other Mg-based alloys and compounds.

## Methods

### Materials synthesis

High purity Mg shots, Bi shots, Sb shots, and Te pieces were weighted according to the composition of $Mg_{3.15}Bi_{1.4975}Sb_{0.5}Te_{0.0025}$ and $Mg_{3.15}Bi_{1.2475}Sb_{0.75}Te_{0.0025}$. Raw materials with a total mass of 12 g were loaded into $ZrO_2$ crucibles in a glove box under Ar atmosphere with $O_2$ and water level below 0.1 ppm. The crucibles were embed into Ta tubes and the Ta tubes were sealed through arc-melting under Ar atmosphere. Then they were put into quartz tubes and sealed in vacuum. $ZrO_2$ crucible can protect Ta tube from the reaction of melting Sb element, and the sealed Ta tube prevent the reaction of Mg vaper with the quartz. The crystal growth follows the procedures, within a rocking furnace the samples were firstly heated up to 1273 K. The molten sample was swung continuously for 2 h in the rocking furnace. Then the samples were slowly cooled down to 1193 K at a rate of 2 K/h and kept for 24 h. Next, the crucible was cooled down to 1073 K at a rate of 2 K/h and annealed for 24 h. In succession, the crucible was cooled down to 973 K at a rate of 2 K/h and maintained for 72 h. Finally, the crucible was cooled down to room temperature naturally.

### Phase and microstructure characterization

Powder XRD was performed using a Bruker D2 Phaser diffractometer with the Cu Kα radiation ($\lambda$ = 1.5418 Å, voltage of 40 kV, and emission current of 40 mA) in Bragg-Brentano geometry. The XRD patterns were collected at room temperature over a 2 theta range of 10–80° with a step size of 0.006° for 40 min. Rietveld refinement against as-collected XRD data for CGBi1.5 was performed using GSAS via the EXPGUI interface[49]. Considering the very small amount of Te doping, a published $Mg_3Bi_2$ structure[27] was modified by accommodating 25% of Sb and 75% of Bi at the *2d* Bi site, *i.e.* with the fixed respective SOF of 0.25 and 0.75; the structure was employed as an initial crystallographic model. A constraint of the same coordinates and thermal displacement factors was then applied to Bi and Sb atoms. During the refinement, isotropic atom thermal displacements were applied to the heavy Sb and Bi atoms, while anisotropic types were set for the relatively light Mg atoms. SOFs of Mg2 and Mg1 were carefully refined, whereas the refinement of the former led to a SOF(Mg2) unchanged from unity,

indicating the unfavored conditions for Mg2 vacancies within the anionic layers. However, a SOF(Mg1) lower than 1.0 was consistently suggested, and the stoichiometry of the sample was finally evaluated. Results of the refinement are available in Supplementary Tables 2 and 3 in the SI file and Fig. 2a in the main manuscript. The microstructures, grain size, and element distribution were conducted by scanning electron microscopy (SEM, S8100, Hitachi) equipped with an energy dispersive spectrometer (EDS) and electron back-scattering diffraction (EBSD) detector. STEM was conducted using JEOL ARM200F.

### Measurement of thermoelectric properties

For high-temperature (>300 K) measurement, both electrical and thermal transport properties were measured at the same direction. The obtained ingots were cut into bars with a dimension of $2.5 \times 2.5 \times 10\ mm^3$ for the simultaneous measurements of Seebeck coefficient and electrical resistivity (Linseis, *LSR-3*, Germany). Disks with diameter of 12.7 mm and thickness of 1.5 mm were cut from the ingots for the thermal diffusivity measurement. The thermal conductivity ($\kappa$) was calculated by $\kappa = \rho C_p D$, where $\rho$ was density measured by Archimedes drainage method, $C_p$ was the specific heat estimated according to the Dulong-Petit law, and $D$ was the thermal diffusivity measured by *LFA 1000* (Linseis, Germany). The measurement uncertainties are 3%, 5%, and 6% for electrical resistivity, Seebeck coefficient, and thermal diffusivity, respectively, which yield an error in $zT$ of ~20%.

For low temperature (150–370 K) measurement, all the transport properties were measured at one bar with a dimension of $3 \times 1.5 \times 7\ mm^3$. The thermal conductivity and Seebeck coefficient were measured simultaneously (Supplementary Fig. 18) using TTMS (Thermal Transport Measurement System, MultiFields Technology), and the electrical resistivity was measured by PPMS (Physical Properties Measurement System, Quantum Design). The bars were polished to $3 \times 0.5 \times 7\ mm^3$ for the Hall coefficient ($R_H$) measurement under a reversible magnetic field ($\pm 2$ T) using PPMS. The Hall carrier concentration ($n_H$) and the Hall carrier mobility ($\mu_H$) were calculated by $n_H = 1/eR_H$ and $\mu_H = \sigma/(en_H)$, respectively, where $e$ is the electronic charge and $\sigma$ is the electrical conductivity.

### Mechanical property measurement

Vickers hardness was tested at room temperature with an applied load of 10 gf maintained for 10 s on a microhardness tester (MH500, China). The compressive strength was performed on samples with dimensions of $2 \times 2 \times 4\ mm^3$ using a universal test machine (Instron 5967, American).

### Module preparation and cooling performance measurement

The sintering process of the materials, interfacial layers and contact layers was shown in our previous work[30]. The test method of contact resistivity was also the same as reported before[30], and the measured contact resistivity in this work was 13.5 $\mu\Omega\ cm^2$ (Supplementary Fig. 17). The obtained sandwich pellets were diced into squares with a dimension of $2.0 \times 2.0 \times 3\ mm^3$ using a dicing machine (DS830, Heyan technology). These legs were polished to $1.3 \times 1.3 \times 3\ mm^3$ to eliminate the influence of surface corrosion layer and then annealed at 613 K for 2 h. The p-type $Bi_2Te_3$ materials were cut into squares with an optimized area of $1.25 \times 1.25\ mm^2$. The fabrication of thermoelectric modules and the cooling performance measurement process can also be found in previous report[30]. The power-generation performance was characterized using Mini-PEM (ADVANCE RIKO, Japan). The hot-side temperature $T_h$ of the module was controlled by a heater, and the cold-side temperature $T_c$ was controlled by the flowing water. Both $T_h$ and $T_c$ were measured by thermocouples of the Mini-PEM. The output power ($P$) and cold-side heat flow ($Q_c$) were recorded using the Mini-PEM. Therefore, the power-generation efficiency was calculated by $\eta = \frac{P}{P+Q_c}$.

### Calculation of weighted mobility

The single parabolic band (SPB) model is expressed as follows:[50]

$$n = \frac{4\pi(2m_d^* k_B T)^{3/2}}{h^3} F_{1/2}(\eta) \tag{1}$$

$$S = \pm \frac{k_B}{e} \left( \frac{(r+5/2)F_{r+3/2}(\eta)}{(r+3/2)F_{r+1/2}(\eta)} - \eta \right) \tag{2}$$

$$L = \frac{k_B^2}{e^2} \frac{(r+3/2)(r+7/2)F_{r+5/2}(\eta)F_{r+1/2}(\eta) - (r+5/2)^2 F_{r+3/2}(\eta)^2}{(r+3/2)^2 F_{r+1/2}(\eta)^2} \tag{3}$$

$$F_n(\eta) = \int_0^\infty \frac{x^n dx}{1+\exp(x-\eta)} \tag{4}$$

In the above equations, $k_B$ is the Boltzmann constant, $e$ the electron charge, $h$ the Plank constant, $r$ the scattering factor, $m_d^*$ the density of state effective mass, $\eta$ the reduced Fermi level, $x$ the reduced carrier energy, and $F_n(\eta)$ the Fermi-Dirac integral.

The weighted mobility $\mu_w$ is calculated by[38]

$$\mu_w = \mu_H \left( \frac{m_d^*}{m_e} \right)^{3/2} \tag{5}$$

where $m_e$ is the electron mass.

### The fitting of grain size and carrier mobility

According to Matthiessen's rule, the total mobility $\mu$ can be depicted as:

$$\mu^{-1} = \mu_{GB}^{-1} + \mu_I^{-1} + \mu_{ph}^{-1} + \mu_{al}^{-1} + \cdots \tag{6}$$

given the carriers are scattered by grain boundary, ionized impurity, acoustic phonons, alloy disorder and so on. Considering that grain size only affects grain-boundary scattering, other terms in Eq. 6 except $\mu_{GB}^{-1}$ can be approximated as constants $C$. Expression of $\mu_{GB}$ is given by:[51]

$$\mu_{GB} = de \left( \frac{1}{2\pi m^* k_B T} \right)^{1/2} \exp\left( \frac{-E_b}{k_B T} \right) \tag{7}$$

where $d$ is the grain size, $m^*$ the band mass, and $E_b$ the height of the potential barrier. The curve of $\mu_H \sim d$ was fitted according to Eqs. 6 and 7. The literature data used for the fitting can be found in Supplementary Table 4.

### Calculation of cooling performance of thermoelectric devices

The cooling capacity ($Q_c$) of a thermoelectric cooling module can be expressed as:

$$Q_c = S_{np} T_c I - \frac{1}{2} I^2 (R+R_c) - K(T_h - T_c) \tag{8}$$

where $S_{np}, T_h, T_c, I, R, R_c, K$ are the total Seebeck coefficient of device, the hot-side temperature of device, the cold-side temperature of device, the electric current, the total electrical resistance of thermoelectric materials on the legs, the total contact resistance from the interface layer, and the thermal conductance through the thermoelectric legs, respectively.

$S_{np}, R, R_c$ and $K$ can be expressed as:

$$S_{np} = N|S_n - S_p| = N \left( \frac{\int_{T_c}^{T_h} S_p(T)dT}{T_h - T_c} - \frac{\int_{T_c}^{T_h} S_n(T)dT}{T_h - T_c} \right) \tag{9}$$

$$R = N\left(\frac{h}{A_n}\rho_n + \frac{h}{A_p}\rho_p\right) = N\left(\frac{h}{A_n}\frac{\int_{T_c}^{T_h}\rho_n(T)dT}{T_h - T_c} + \frac{h}{A_p}\frac{\int_{T_c}^{T_h}\rho_p(T)dT}{T_h - T_c}\right) \quad (10)$$

$$R_c = 2N\left(\frac{\rho_{nc}}{A_n} + \frac{\rho_{pc}}{A_p}\right) \quad (11)$$

$$K = 2N\left(\frac{A_n}{h}\kappa_n + \frac{A_p}{h}\kappa_p\right) = N\left(\frac{A_n}{h}\frac{\int_{T_c}^{T_h}\kappa_n(T)dT}{T_h - T_c} + \frac{A_p}{h}\frac{\int_{T_c}^{T_h}\kappa_p(T)dT}{T_h - T_c}\right) \quad (12)$$

where $N$, $S_n$, $S_p$, $h$, $A_n$, $A_p$, $\rho_n$, $\rho_p$, $\rho_{nc}$, $\rho_{pc}$, $\kappa_n$, $\kappa_p$ are the number of the thermocouples, the Seebeck coefficient of n-type and p-type materials, the height of the thermoelectric legs, the cross-section area of n-type and p-type legs, the electrical resistivity of n-type and p-type materials, the contact resistivity of n-type and p-type legs, the thermal conductivity of n-type and p-type thermoelectric materials, respectively.

The input power ($p$) on the device can be expressed as:

$$p = I^2(R + R_c) + S_{np}(T_h - T_c)I \quad (13)$$

The COP is defined as the ratio of $Q_c$ to $p$:

$$COP = \frac{S_{np}T_c I - \frac{1}{2}I^2(R + R_c) - K(T_h - T_c)}{I^2(R + R_c) + S_{np}(T_h - T_c)I} \quad (14)$$

The cooling power density ($q_c$) is given by:

$$q_c = \frac{S_{np}T_c I - \frac{1}{2}I^2(R + R_c) - K(T_h - T_c)}{S_{ceramic}} \quad (15)$$

where $S_{ceramic}$ is the area of ceramic plate.

## Data availability

All data generated or analyzed in this study are included in the published article and its Supplementary Materials. The data will be made available on request.

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

## Acknowledgements

This work is supported by the National Key Research and Development Program of China (Grant no. 2022YFB3803900 H.-Z.Z., 2018YFA0702100 H.-Z.Z., and 2021YFA0718700 G.L.) and the National Natural Science Foundation of China (Grant no. 52172262 H.Z. and 52172259 G.L.).

## Author contributions

H.-Z.Z., H.Z., and N.C. discussed and designed the experiments. N.C. synthesized the samples. N.C. and Y.Y. performed the microstructure characterizations. Z.F. conducted the crystallographic structure analysis and writing (review & editing). N.C. and G.L. measured the thermo-electric properties. N.C. assembled the modules. X.Z., J.Y., and T.L. helped with the measurements of device performance and theoretical stimulation. Q.L., X.W., and Y.S. were involved in the discussion of sample synthesis. N.C., H.-Z.Z., and H.Z. wrote the manuscript. All authors contributed to the data analysis and edited the manuscript.

## Competing interests

The authors declare no competing interests.
