## [Peer Review File · Nature Communications]

Improved Figure of Merit (z) at Low Temperatures for Superior Thermoelectric Cooling in $\text{Mg}_3(\text{Bi,Sb})_2$REVIEWER COMMENTS

Reviewer #1 (Remarks to the Author):

In this work, the authors reported the thermoelectric properties of Mg₃Bi₂-based device with the largest temperature difference of 68-82 K. The higher mobility due to the larger grain size leads to higher z and better cooling performance. Even though similar zT values and idea of larger grain itself can be found in previous reports, this study provides new practical way to improve Mg₃Bi₂-based thermoelectric cooling device. Mg₃Bi₂ is a hot topic in the field of thermoelectrics and the results of this work are important for the development of the Mg₃Bi₂ based module. Since the work is very complete from material synthesis/measurements to device fabrication/evaluation, I recommend publishing this work in nature communications. Some comments for authors are listed below.

1. Why Mg₂Cu was used as the contact layer? Does Cu or Mg react or migrate into the thermoelectric leg after long term testing? Both Cu and Mg can have ionic conduction by operating the device. The previous work reported other elements can be a good candidate for the contact layer. Is there any possibility of further improvement with other composition for the contact layer?
2. Any mechanical property measurements for the coarse grained samples? Sometimes, the larger grain samples have lower mechanical strength due to less defects. Mechanical robustness is also important for the thermoelectric cooling device and power generation.
3. As authors mentioned in the last paragraph, stability of the Mg₃Bi₂ module is very important. What is the measured condition for long-term testing? (Under vacuum? Is it a continuous measurement with current flow?) They attribute the better stability of module less Mg vacancy in figure 4f, but it might just be due to low temperature and ideal measurement condition without air and water moisture.

Reviewer #2 (Remarks to the Author):

The manuscript provided a facile growth of coarse-grain Mg₃Bi₂-xSb_x polycrystals with a high thermoelectric performance through the nested-crucible approach. High performance thermoelectric cooling and power generation modules were successfully fabricated, representing the application prospects of Mg₃(Bi,Sb)₂-based materials at near-room temperature. This study is interesting and I recommend publishing after a minor revision.

1. There are some writing errors. For example, "...through the carrier concentration (n) and the band effective mass (m^*) (ref. 3)...." in the first paragraph on page 3 and "range and slightly higher than that of $89\pm 2\%$ as reported for Mg_{3.2}Sb_{1.5}Bi_{0.49}Te_{0.01} (ref. 39), ..." on page 8. Both of the "ref." should be removed. The authors should go through the manuscript again.
2. According to the manuscript, testing bars and the thermoelectric unit leg were directly cut from the ingots obtained by the approach of coarse grain growth. As known, the density has a great impact on the thermoelectric and mechanical performance. The authors should provide the density of the synthesized ingots.
3. The authors claimed that "... Moreover, the as-grown coarse-grained bulk crystals (with a dimension of $\Phi 12.7$ mm \times 13 mm) also exhibit excellent uniformity and mechanical performance, ...". Due to the lack of data, more supporting evidence are necessary.
4. According to Fig.4a, the authors claimed that "..., and also comparable to the state-of-the-art commercial Bi₂Te₃ cooling module....". However, this statement makes readers believe that the maximum ΔT of CGBi_{1.5}/Bi_{0.5}Sb_{1.5}Te₃ based module is superior to that of current

commercial Bi₂Te₃ cooling module. To make it more convincing, the authors should add more data from well-known companies such as KELK, Ferrotec, and so on.

5. The authors claimed that "...Fig. S7 reveals the high reproducibility of this synthesis method represented by thermoelectric properties of several batches of CGBi_{1.5} samples...". However, the provided data is difficult to convince. The author should give more discussion on it.

6. In Fig. 4a and 4e, the testing temperatures of the CGBi_{1.5}/Bi_{0.5}Sb_{1.5}Te₃ are much lower than that of the CGBi_{1.25}/Bi_{0.5}Sb_{1.5}Te₃. The authors should provide reasonable explanation.

7. The authors claimed that "..., the electrical resistivity remained unchanged over 16 months, being the best record ever, ...". As we know, aging conditions have a serious affect upon the stability of thermoelectric materials and devices. Thus, the details of aging testing should be provided. In addition, the electrical resistivity of the CGBi_{1.5} slightly decrease as aging progresses. As known, the generation of Mg vacancies during aging will cause an increase of the electrical resistivity for n-type CGBi_{1.5}. The authors should give some explanation on it.

Reviewer #3 (Remarks to the Author):

Designing good thermoelectric materials and devices involves careful balancing of thermal and electronic transport properties. For certain classes of materials with intrinsically low thermal conductivity, a facile way to improve their thermoelectric performance is by preparing coarse grained samples to improve the electronic transport (carrier mobility). However, although this sounds simple in theory, preparing coarse-grained materials often comes with unexpected complications, especially when it involves volatile starting elements such as Mg₂Bi_{2-x}Sb_x, MgAgSb, and other Zintl phases.

In this work, the authors report on a strategy to prepare coarse-grained Mg₃Bi_{2-x}Sb_x samples. A three-layer nested-crucible consisting of ZrO₂/Ta/SiO₂ was used to seal the highly reactive and corrosive Mg and Sb. As a result, samples with grain size of up to 0.8 mm were successfully synthesized, with correspondingly high carrier mobility and figure of merit z . In particular, the weighted-mobility reported in this work is the record-breaking amongst Mg₃Bi_{2-x}Sb_x compounds. Structural characterization such as XRD and high resolution STEM were employed to support the conclusion. In addition, both power generation and cooling performance of the thermoelectric device based on the coarse grained sample were reported.

Overall, this is a complete work, with meaningful objective, and impactful results. Importantly, I can envisage the strategy reported here to be applied to prepare coarse-grained samples of other high performance thermoelectrics that are otherwise difficult to achieve. I would recommend publication after the following points are addressed:

Major points:

- The temperature profiles need to be elaborated further. For example, why is there a need for 2-step cooling (1073K and 973K) and the reason behind holding for 24h and 72h. A simple rationale behind this will be helpful since this is one of the main novelties in the paper.
- The cooling dT of 68K is truly remarkable, how does this value compare to the theoretical estimation of $dT_{max} = [(z)(T_c)^2]/2$?

- In Figure 3b on Hall mobility vs grain size, since the fitting curve is based on a physical model in equation S6 and S7, the values used for the fitting parameters can be clarified in the manuscript.
- The details and equation used for the prediction curve for COP and Q_c in Figure 4b and 4c can be included in the supporting information.
- When discussing Q_c max in figure 4b, it is helpful to state in terms of W/cm^2 for easier comparison with modules having different sizes.
- It is claimed that the electrical resistivity is stable after 16 months, which is very stable. Is there any accelerated way for such testing? Do we have to wait for 16 months for testing such stability?

Minor points:

- Is the $ZrO_2/Ta/SiO_2$ three-layer nested-crucible commercially available? If not, more description on this will be helpful for others who are interested to use similar strategy.
- The yellow highlights throughout the manuscript maybe misunderstood, please remove them in the final version.
- The sentence in the introduction "... with grain size up to 1.0 mm, which approaches two thirds of a thermoelectric unit leg in the devices." Can be improved for clarity. I would suggest "... with grain size up to 1.0 mm. Such grain size encompasses almost two-third of the length of a thermoelectric leg in the devices"
- "Then kept swinging for 2 h to allow the complete reaction of Mg/Bi/Sb and the formation of a uniform liquid phase." Can be : "The molten sample was swung continuously for 2 h in the rocking furnace to allow for the complete reaction of Mg/Bi/Sb and the formation of a uniform liquid phase."
- The max dT during cooling is likely an underestimation, since it seems that the fill factor and aspect ratio have not yet been optimized in the 7-pair device.
- "high reproductivity" should be "high reproducibility"
- Inset in Fig 1c, 3d are too small and low resolution.

Overall, I think this work is of high quality and novelty and would like to recommend acceptance after the aforementioned points are addressed, especially the methodology.

Response to Reviewer Reports

All changes to the manuscript and supporting information are indicated in blue text below and highlighted in the respective documents.

Reviewer #1:

In this work, the authors reported the thermoelectric properties of Mg_3Bi_2 -based device with the largest temperature difference of 68-82 K. The higher mobility due to the larger grain size leads to higher z and better cooling performance. Even though similar zT values and idea of larger grain itself can be found in previous reports, this study provides new practical way to improve Mg_3Bi_2 -based thermoelectric cooling device. Mg_3Bi_2 is a hot topic in the field of thermoelectrics and the results of this work are important for the development of the Mg_3Bi_2 based module. Since the work is very complete from material synthesis/measurements to device fabrication/evaluation, I recommend publishing this work in nature communications. Some comments for authors are listed below.

Response: We are grateful to the reviewer for her/his thoughtful analysis of our manuscript and for the generous and positive comments. We consider the specific points raised below.

1. Why Mg_2Cu was used as the contact layer? Does Cu or Mg react or migrate into the thermoelectric leg after long term testing? Both Cu and Mg can have ionic conduction by operating the device. The previous work reported other elements can be a good candidate for the contact layer. Is there any possibility of further improvement with other composition for the contact layer?

Response: The reliable stabilities between the contact layer and the thermoelectric leg are clearly vital characteristics. We completely agree with the reviewer that this is a topic that requires careful consideration.

Although the reviewer may not notice it, it had been introduced in our original manuscript briefly that the design of Mg_2Cu contact layer with outstanding chemical,

physical and thermal stabilities is the contribution to applications of $\text{Mg}_3(\text{Sb,Bi})_2$ -based TE modules from our previous work (*Joule*, **2022**, 6, 193-204). For the first two questions above, the comprehensive characterization and testing results from our previous work provided details, enabling us readily providing concise information as below:

(i) The thermodynamic stability of the $\text{Mg}_2\text{Cu}/\text{Mg}_3\text{Bi}_{1.5}\text{Sb}_{0.5}$ interface can be confirmed by the ternary phase diagram of $\text{Mg}-\text{Mg}_2\text{Cu}-\text{Mg}_3\text{Bi}_{1.5}\text{Sb}_{0.5}$ (Fig. R1). No new phases can be formed at the interface of $\text{Mg}_2\text{Cu}/\text{Mg}_3\text{Bi}_{1.5}\text{Sb}_{0.5}$ even with high temperature annealing (at 703 K).

(ii) It has been reported that Fe and Ni can be used as contact materials, where Fe/Ni powders are directly added on top of the $\text{Mg}_3(\text{Bi,Sb})_2$ powder for an one-step sintering treatment at high temperatures (up to 1053 K) to obtain a low contact resistance¹. Contrastively, the Mg_2Cu contact layer can be prepared by sintering Mg_2Cu powders with bulk $\text{Mg}_3(\text{Bi,Sb})_2$, which is a suitable choice for coarse-grained (i.e. in our current work) and single crystal samples. Importantly, the Mg_2Cu contact layer could be sintered at a relative low temperature of 803 K, which helps preventing obvious performance deteriorations of $\text{Mg}_3(\text{Bi,Sb})_2$ caused by any further loss of Mg (*cf.* that during the high temperature sintering of 1053 K for Fe/Ni).

(iii) Our previous work shows that after the sintering process, a clear $\text{Mg}_2\text{Cu}/\text{Mg}_3\text{Bi}_{1.5}\text{Sb}_{0.5}$ interface is observed and the obtained thin diffusion layer of $\sim 0.428 \mu\text{m}$ at the interface affirms the good chemical and thermal stabilities between Mg_2Cu and $\text{Mg}_3(\text{Bi,Sb})_2$.

(iv) Mg_2Cu has a similar coefficient of thermal expansion (CTE) with that of $\text{Mg}_3(\text{Bi,Sb})_2$, ensuring the reduction of thermal stress in devices during operation.

Regarding the 3rd question the reviewer raised, we had indeed been in mind of previous and current state of the art of both elemental and compound contact layer candidates.

We also compared the aging time dependence of contact resistivity of materials with Mg_2Cu and Fe (one of the reported high-performance candidate) as contact layers, respectively. The contact resistivity of $\text{Cu}/\text{Mg}_2\text{Cu}/\text{Mg}_{3.2}\text{Bi}_{1.4975}\text{Sb}_{0.5}\text{Te}_{0.0025}$ multi-layers

remains unchanged after annealing at 703 K for 150 h, while that of $\text{Mg}_{3.2}\text{Sb}_{1.5}\text{Bi}_{0.49}\text{Te}_{0.01}/\text{Fe}$ increases significantly with annealing time at 523 K. The constant contact resistivity during annealing indicates the great thermal stability of the Mg_2Cu based contact layer, which shows remarkable advantages in reliability over that of Fe. In addition, the sufficient Mg in Mg_2Cu will serve as the Mg source for $\text{Mg}_3(\text{Bi,Sb})_2$ which helps to stabilize the n-type carrier transport behavior.

Fig. R1. Mg-Cu-Sb ternary phase diagrams at room temperature.

Considering above, Mg_2Cu demonstrates traits of good stability, low contact resistance, and simple preparation process, which can effectively meet the needs at the device fabrication level. We do, however, share the same opinion with the reviewer that the performance of thermoelectric device can be largely promoted with further improved contact resistivity. We propose that this can be potentially achieved through materials designs and the manipulation of interface roughness by advanced film preparations etc. We are exploring such further improvements, which might suit better for a separate study of another day.

2. Any mechanical property measurements for the coarse grained samples? Sometimes, the larger grain samples have lower mechanical strength due to less defects. Mechanical robustness is also important for the thermoelectric cooling device and power generation.

Response: The mechanical robustness is another vital criterion for device-level applications, and we are very grateful to the reviewer for her/his professional perspective. We can also acknowledge that the inclusion of mechanical properties not only improves how our work is presented, but also is instructive towards the potential large-scale applications of our fabricated devices.

Thus, we have performed additional Vickers hardness and compressive strength measurements on the CGBi1.5 materials, where the excellent mechanical properties can be confirmed.

Motivated by the comment, we have added more corresponding discussion in the manuscript and supplementary information as below:

Moreover, the as-grown CGBi1.5 bulk materials show a Vickers hardness of 0.54 GPa, close to that of 0.61 GPa for $\text{Mg}_{3.065}\text{Sb}_{1.3}\text{Bi}_{0.7}\text{Gd}_{0.015}$ as reported in the literature². The compressive strength of the CGBi1.5 sample is greater than 300 MPa, because no cracks occur under the loading stress of 300 MPa, which is consistent with the literature results², as shown in Fig. S3. The above results affirm that excellent mechanical properties of our coarse-grained materials, fulfilling the needs for TE device manufacturing.

Fig. S3. The compressive stress-strain curves of CGBi1.5 bulk materials.

3. As authors mentioned in the last paragraph, stability of the Mg_3Bi_2 module is very

important. What is the measured condition for long-term testing? (Under vacuum? Is it a continuous measurement with current flow?) They attribute the better stability of module less Mg vacancy in figure 4f, but it might just be due to low temperature and ideal measurement condition without air and water moisture.

Response: We thank the reviewer for the rigorous and careful scrutiny very much. The module stability is one other important criterion for device applications, and we are very happy to respond to the reviewer's comment.

First of all, we expect to develop the packaging scheme for the devices to achieve a long service time. The coarse-grained bars ($\sim 1.7 \times 2.7 \times 10 \text{ mm}^3$) were placed in the glove box filled with Ar (O_2 and H_2O levels generally lower than 0.5 ppm) and taken out every week for room-temperature electrical property tests. The tests were performed in a *LSR-3* equipment under a helium atmosphere. We acknowledge that short air exposures cannot be avoided when mounting the sample in *LSR-3*.

The stability of the sample investigated in the work suggests that the long-term service of the materials to be realized by device packaging technique in the future.

For the chemical stability of $\text{Mg}_3(\text{Bi,Sb})_2$ -based materials, we believe that the internal Mg vacancy and external storage conditions are both important. The atom diffuses essentially faster (more than one magnitude) in the lattice with high content of vacancy than in perfect crystal. Therefore, the diffusion of Mg is remarkably suppressed in the coarse-grained sample with significantly reduced Mg vacancy, which decelerates the consuming process of Mg and thus prolongs the service time of device.

Reviewer #2:

The manuscript provided a facile growth of coarse-grain $\text{Mg}_3\text{Bi}_{2-x}\text{Sb}_x$ polycrystals with a high thermoelectric performance through the nested-crucible approach. High performance thermoelectric cooling and power generation modules were successfully fabricated, representing the application prospects of $\text{Mg}_3(\text{Bi,Sb})_2$ -based materials at near-room temperature. This study is interesting and I recommend publishing after a minor revision.

Response: We thank the reviewer for her/his supportive comments and the suggested improvements. We consider these below.

1. There are some writing errors. For example, "...through the carrier concentration (n) and the band effective mass (m^*) (ref. 3)..." in the first paragraph on page 3 and "range and slightly higher than that of $89\pm 2\%$ as reported for $\text{Mg}_{3.2}\text{Sb}_{1.5}\text{Bi}_{0.49}\text{Te}_{0.01}$ (ref. 39), ..." on page 8. Both of the "ref." should be removed. The authors should go through the manuscript again.

Response: We are thankful to the reviewer for pointing out the writing errors. The citations above have been reformatted. We have also checked through the entire manuscript and supplementary materials carefully, all similar issues have been corrected and marked in the revised files.

2. According to the manuscript, testing bars and the thermoelectric unit leg were directly cut from the ingots obtained by the approach of coarse grain growth. As known, the density has a great impact on the thermoelectric and mechanical performance. The authors should provide the density of the synthesized ingots.

Response: We are highly appreciative to the reviewer for highlighting this know vital issue; and we completely agree. We had tested the density of bulk materials in the original manuscript, which however was not explicitly described. As suggested by the reviewer, we have performed additional density measurements. Together with previous results, it can be confirmed that the synthesized ingots consistently have high density values in our work.

In order to improve the clarity, the above relevant contents have been added in the revised manuscript and supplementary materials (as Table S1) more explicitly as below: The measured densities of as-grown CGBi1.5 and CGBi1.25 materials are 5.36 g/cm³ and 5.14 g/cm³, respectively, agreeing well with the experimental and theoretical values^{3,4} (Table S1).

Table. S1. Densities of as-grown CGBi1.5 and CGBi1.25 samples, and the calculated densities from the two end members assuming a linear relation⁴.

Samples	CGBi1.5		CGBi1.25	
	Measured	Calculated	Measured	Calculated
Density (g·cm ⁻³)	5.36	5.39	5.14	5.16

3.The authors claimed that “... Moreover, the as-grown coarse-grained bulk crystals (with a dimension of Φ 12.7 mm \times 13 mm) also exhibit excellent uniformity and mechanical performance, ...”. Due to the lack of data, more supporting evidence are necessary.

Response: We thank the reviewer for the rigorous comment on the uniformity and mechanical performance. For the former, the composition uniformity can be readily confirmed by the EDS spectra and mapping results, as evidenced in Fig. 2b and Fig. S6. Herein, we can also provide more evidence on the uniform thermoelectric performance of difference parts of (a) one ingot sample and (b) from different ingots (*i.e.* Figures S9 and S10 below). The corresponding results are included in the revised manuscript and supplementary materials as follows:

The electrical transport properties of CGBi1.5 samples in different parts and from different samples have been tested, (Fig. S9 and S10), attesting to the uniform thermoelectric performance achieved through our method.

Fig. S9. The electrical transport properties as a function of temperature along the out-of-plane and in-plane directions in CGBi_{1.5} bulk samples from three different batches. (a) Electrical resistivity, (b) Seebeck coefficient and (c) power factor. The almost identical experimental data in two directions not only shows the isotropic properties of the coarse grain materials, but also indicates the uniformity of the samples.

Fig. S10. Comparison of the electrical transport properties among different parts of the as-grown CGBi_{1.5} bulk ingot. (a) Electrical resistivity, (b) Seebeck coefficient and (c) power factor. Four bars with dimensions of $\sim 2.5 \times 2.5 \times 8 \text{ mm}^3$ were cut in the out-of-plane direction from one CGBi_{1.5} bulk materials. No significant difference in the electrical transport properties of the four bars was observed.

For the latter, *i.e.* the mechanical performance, Reviewer 1 also asks the same question (Above Comment 2). To expand further on our previous comment about without wishing to be repetitive, we have performed additional experiments and added the corresponding discussion in the revised manuscript and supplementary information as below:

Moreover, the as-grown CGBi_{1.5} bulk materials show a Vickers hardness of 0.54 GPa, close to that of 0.61 GPa for Mg_{3.065}Sb_{1.3}Bi_{0.7}Gd_{0.015} as reported in the literature². The compressive strength of the CGBi_{1.5} sample is greater than 300 MPa, because no cracks occur under the loading stress of 300 MPa, which is consistent with the literature

results², as shown in Fig. S3. The obtained coarse-grained materials have excellent mechanical properties and meet the needs of device manufacturing.

Fig. S3. The compressive stress-strain curves of CGBi1.5 bulk materials.

4. According to Fig. 4a, the authors claimed that "..., and also comparable to the state-of-the-art commercial Bi₂Te₃ cooling module....". However, this statement makes readers believe that the maximum ΔT of CGBi1.5/Bi_{0.5}Sb_{1.5}Te₃ based module is superior to that of current commercial Bi₂Te₃ cooling module. To make it more convincing, the authors should add more data from well-known companies such as KELK, Ferrotec, and so on.

Response: We are grateful to the reviewer for the rigorousness and apologize for the ambiguity in the statement. We note at the outset, that we did not state in the original manuscript that the maximum ΔT of CGBi1.5/Bi_{0.5}Sb_{1.5}Te₃ based module is superior to that of current commercial Bi₂Te₃ cooling module. Motivated by the reviewer, we have revisited current art of commercial cooling devices from some well-known companies, and the maximum ΔT of have been included in the updated Fig. 4a for clear comparisons in the revised manuscript (also shown as Figure R2 below).

Fig. R2. The comparison of maximum cooling temperature differences between $Mg_3Bi_{2-x}Sb_x/Bi_{0.5}Sb_{1.5}Te_3$ ($x=0.5, 0.75$) TE modules and commercial Bi_2Te_3 -based module with a variety of hot-side temperatures T_h .

5. The authors claimed that “...Fig. S7 reveals the high reproducibility of this synthesis method represented by thermoelectric properties of several batches of CGBi1.5 samples...”. However, the provided data is difficult to convince. The author should give more discussion on it.

Response: Many thanks to the reviewer for raising this point. Firstly, the electrical resistivity, Seebeck coefficient, and thermal diffusivity measurements have associated uncertainties of 3%, 5%, and 6%, respectively, resulting in an error in the figure of merit (zT) of approximately 20%. Secondly, small variations in carrier concentration are inevitable among CGBi1.5 bulk materials grown from different batches, although they exhibit similar power factors (PF). The difference between the maximum and minimum PF values across different batches is within 7.5% at room temperature. Furthermore, it is worth noting that higher power factor is often accompanied with increased thermal conductivity. As a result, the difference in zT values at room temperature is limited to within 2.3%. In light of the points mentioned above, this sentence is reversed as follow: “...Fig. S11 reveals the good reproducibility of this synthesis method represented by

thermoelectric properties of several batches of CGBi1.5 samples....”

Fig. S11. Thermoelectric properties of samples of CGBi1.5 from different batches in this work demonstrate the good reproducibility. (a) Electrical resistivity, (b) Seebeck coefficient, (c) power factor, (d) thermal conductivity and (e) zT .

6. In Fig. 4a and 4e, the testing temperatures of the CGBi1.5/Bi_{0.5}Sb_{1.5}Te₃ are much lower than that of the CGBi1.25/Bi_{0.5}Sb_{1.5}Te₃. The authors should provide reasonable explanation.

Response: We are grateful to the reviewer for this question. Due to the larger bandgap of CGBi1.25 bulk materials, the bipolar effect is effectively suppressed, leading to the occurrence of peak zT values at higher temperatures compared to CGBi1.5 samples. As illustrated in Fig. R3, the peak zT values for CGBi1.5 and CGBi1.25 materials locates at ~ 375 K and ~ 475 K, respectively. In both thermoelectric cooling and power generation devices, higher zT value correspond to higher energy conversion efficiency when the temperatures of the hot and cold sides are fixed. Since CGBi1.25 material exhibits higher zT values at elevated temperature, device based on this material demonstrates superior performance at higher hot-side temperature. Therefore, we increase the test temperature for CGBi1.25/Bi_{0.5}Sb_{1.5}Te₃ device.

Fig. R3. Comparison of zT between CGBi1.5 and CGBi1.25 samples.

7. The authors claimed that “..., the electrical resistivity remained unchanged over 16 months, being the best record ever, ...”. As we known, aging conditions have a serious affect upon the stability of thermoelectric materials and devices. Thus, the details of aging testing should be provided. In addition, the electrical resistivity of the CGBi1.5 slightly decrease as aging progresses. As known, the generation of Mg vacancies during aging will cause an increase of the electrical resistivity for n-type CGBi1.5. The authors should give some explanation on it.

Response: We thank the reviewer for raising this point; the stability of materials is also very important for device applications. We stored the coarse-grained bars ($\sim 1.7 \times 2.7 \times 10 \text{ mm}^3$) in a glove box and tested the room-temperature electrical transport property every week. The measurements were performed using an *LSR-3* under a helium atmosphere. The electrical resistivity measurement had an uncertainty of 3%, and the resistivity values depicted in Fig. 4f fluctuated within a range of $\pm 3\%$. Based on this observation, we conclude that the electrical resistivity remained nearly unchanged.

In this revised version, we have made modifications to Fig. 4f and the corresponding discussion as below:

Lastly, regarding the stability of $\text{Mg}_3\text{Bi}_{2-x}\text{Sb}_x$ materials, we observed that with reduced Mg1 vacancy, the electrical durability of coarse-grained CGBi1.5 bulk materials stored in the glove box was significantly improved, as shown in Fig. 4f.

Fig. R4. Aging time dependence of electrical resistivity of CGBi1.5 samples.

Reviewer #3:

Designing good thermoelectric materials and devices involves careful balancing of thermal and electronic transport properties. For certain classes of materials with intrinsically low thermal conductivity, a facile way to improve their thermoelectric performance is by preparing coarse grained samples to improve the electronic transport (carrier mobility). However, although this sounds simple in theory, preparing coarse-grained materials often comes with unexpected complications, especially when it involves volatile starting elements such as $\text{Mg}_2\text{Bi}_{2-x}\text{Sb}_x$, MgAgSb , and other Zintl phases.

In this work, the authors report on a strategy to prepare coarse-grained $\text{Mg}_3\text{Bi}_{2-x}\text{Sb}_x$ samples. A three-layer nested-crucible consisting of $\text{ZrO}_2/\text{Ta}/\text{SiO}_2$ was used to seal the highly reactive and corrosive Mg and Sb. As a result, samples with grain size of up to 0.8 mm were successfully synthesized, with correspondingly high carrier mobility and figure of merit z . In particular, the weighted-mobility reported in this work is the record-breaking amongst $\text{Mg}_3\text{Bi}_{2-x}\text{Sb}_x$ compounds. Structural characterization such as XRD and high resolution STEM were employed to support the conclusion. In addition, both power generation and cooling performance of the thermoelectric device based on the coarse grained sample were reported.

Overall, this is a complete work, with meaningful objective, and impactful results. Importantly, I can envisage the strategy reported here to be applied to prepare coarse-grained samples of other high performance thermoelectrics that are otherwise difficult to achieve. I would recommend publication after the following points are addressed:

Response: We thank the reviewer for the positive comments. We are grateful for her/his suggested modifications, which have allowed us to improve the paper significantly. We have addressed your comments below in detail.

Major points:

1. The temperature profiles need to be elaborated further. For example, why is there a need for 2-step cooling (1073K and 973K) and the reason behind holding for 24h and 72h. A simple rationale behind this will be helpful since this is one of the main novelties

in the paper.

Response: We completely agree with the reviewer that this is a critical section that needs to be covered in detail. We have added detailed discussion about the temperature profiles in the manuscript as follows:

The crystal growth process involved the following steps: The samples were placed in a rocking furnace and initially heated to 1273 K to melt the precursors. The molten sample was continuously swung for 2 h to ensure complete reaction and the formation of a uniform liquid phase. Subsequently, the samples were slowly cooled down to 1193 K at a rate of 2 K/h and held at this temperature for 24 h. At 1193 K, the high-temperature $\text{Mg}_3(\text{Bi,Sb})_2$ phase began to solidify from the melt state. Nucleation likely initiated at specific sites in the crucible, such as the junction of the sidewall and the bottom of the crucible. The 24-hour holding period facilitated the preservation of high-quality nuclei for subsequent grain growth, while low-quality nuclei gradually shrank and melted away. Next, the crucible was further cooled down to 1073 K at a rate of 2 K/h and annealed for 24 h. At this stage, the amount of high-temperature phase (cubic phase) gradually increased by consuming the liquid phase during cooling. The small amount of remaining liquid phase between the grains promoted grain growth during the annealing process, as depicted in the inset of Fig. R5. Following this, the crucible was cooled down to 973 K at a rate of 2 K/h and held at this temperature for 72 h. Throughout this step, the high-temperature phase completely transformed into the uniform hexagonal phase (room-temperature phase) of $\text{Mg}_3\text{Bi}_{2-x}\text{Sb}_x$ ($x = 0.5, 0.75$). The longtime annealing helped eliminate defects in the material. Finally, the crucible was allowed to naturally cool down to room temperature.

Fig. R5. The schematic of the pseudo-binary Mg-Bi_{0.75}Sb_{0.25} phase diagram and the synthesis procedures.

2. The cooling dT of 68K is truly remarkable, how does this value compare to the theoretical estimation of $dT_{\max} = [(z)(T_c)^2]/2$?

Response: We thank the reviewer for raising this point. Based on the equation $\Delta T_{\max} = \frac{1}{2}zT_c^2$, when the hot-side temperature (T_h) of the CGBi_{1.5}/Bi_{0.5}Sb_{1.5}Te₃ cooling module is maintained at 300 K, the cold-side temperature (T_c) can be cooled to 227 K, resulting in a theoretical estimation of ΔT_{\max} as 73 K. The average z value of $2.85 \times 10^{-3} \text{ K}^{-1}$ between 227 K and 300 K was utilized for this prediction. Our measurement data (68 K) reached 93% of the theoretical estimation. The difference between the measured and theoretical values may be attributed to several factors, including inevitable electrical and thermal losses at the interface, deviations in the thermoelectric performance of the TE legs compared to the bulk materials, and errors in temperature measurement.

3. In Figure 3b on Hall mobility vs grain size, since the fitting curve is based on a

physical model in equation S6 and S7, the values used for the fitting parameters can be clarified in the manuscript.

Response: Many thanks to the reviewers for this suggestion. According to Equations S6 and S7, we assumed that the grain growth has nearly no effect on other scattering sources except the grain boundary. Consequently, the other terms in Equation S6, except for μ_{GB}^{-1} , can be approximated as an independent constant denoted as C. In the case of the $Mg_3(Bi,Sb)_2$ system with a fixed Bi/Sb ratio, the band structure (represented by effective mass m^*) remains unchanged. The value of m^* ($0.36 m_e$) was determined using the SPB model. Thus, the only unknown variable in Equations S6 and S7 is the height of the potential barrier, E_b . We collected literature data of grain sizes (d) and their corresponding carrier mobilities (μ_H) for Bi-rich n-type $Mg_3(Bi,Sb)_2$ materials. By fitting the numerical relation of $\mu_H \sim d$, we determined a value of 170 meV for E_b .

The literature data used for the fitting process are provided in the supplementary materials (Table S4) as follows:

Table S4. The fitting parameters used in the $\mu_H \sim d$ curve. m^* of $0.36 m_e$ was used for fitting.

Grain sizes (d , μm)	Carrier mobility (μ_H , $cm^2 \cdot V^{-1} \cdot s^{-1}$)	Reference
5	78	
10	133	5
20	172	
50	169	6
9000	220	3
25	165	This work, SPS sample
800	210	This work, CGBi1.5

4. The details and equation used for the prediction curve for COP and Q_c in Figure 4b and 4c can be included in the supporting information.

Response: The reviewer is correct that we should provide more details about the

prediction of thermoelectric device performance. Thank you for your suggestions. The details and equations for the prediction have been added to the supplementary materials as follows:

Calculation of cooling performance of thermoelectric devices:

The cooling capacity (Q_c) of a thermoelectric cooling module can be expressed as:

$$Q_c = S_{np}T_c I - \frac{1}{2}I^2(R + R_c) - K(T_h - T_c) \quad (S8)$$

where S_{np} , T_h , T_c , I , R , R_c , K are the total Seebeck coefficient of device, the hot-side temperature of device, the cold-side temperature of device, the electric current, the total electrical resistance of thermoelectric materials on the legs, the total contact resistance from the interface layer, and the thermal conductance through the thermoelectric legs, respectively.

S_{np} , R , R_c and K can be expressed as:

$$S_{np} = N|S_n - S_p| = N \left(\frac{\int_{T_c}^{T_h} S_p(T) dT}{T_h - T_c} - \frac{\int_{T_c}^{T_h} S_n(T) dT}{T_h - T_c} \right) \quad (S9)$$

$$R = N \left(\frac{h}{A_n} \rho_n + \frac{h}{A_p} \rho_p \right) = N \left(\frac{h}{A_n} \frac{\int_{T_c}^{T_h} \rho_n(T) dT}{T_h - T_c} + \frac{h}{A_p} \frac{\int_{T_c}^{T_h} \rho_p(T) dT}{T_h - T_c} \right) \quad (S10)$$

$$R_c = 2N \left(\frac{\rho_{nc}}{A_n} + \frac{\rho_{pc}}{A_p} \right) \quad (S11)$$

$$K = 2N \left(\frac{A_n}{h} \kappa_n + \frac{A_p}{h} \kappa_p \right) = N \left(\frac{A_n}{h} \frac{\int_{T_c}^{T_h} \kappa_n(T) dT}{T_h - T_c} + \frac{A_p}{h} \frac{\int_{T_c}^{T_h} \kappa_p(T) dT}{T_h - T_c} \right) \quad (S12)$$

where N , S_n , S_p , h , A_n , A_p , ρ_n , ρ_p , ρ_{nc} , ρ_{pc} , κ_n , κ_p are the number of the thermocouples, the Seebeck coefficient of n-type and p-type materials, the height of the thermoelectric legs, the cross-section area of n-type and p-type legs, the electrical resistivity of n-type and p-type materials, the contact resistivity of n-type and p-type legs, the thermal conductivity of n-type and p-type thermoelectric materials, respectively.

The input power (p) on the device can be expressed as:

$$p = I^2(R + R_c) + S_{np}(T_h - T_c)I \quad (S13)$$

The coefficient of performance (COP) is defined as the ratio of Q_c to p :

$$\text{COP} = \frac{S_{\text{np}}T_c I - \frac{1}{2}I^2(R+R_c) - K(T_h - T_c)}{I^2(R+R_c) + S_{\text{np}}(T_h - T_c)I} \quad (\text{S14})$$

The cooling power density (q_c) is given by:

$$q_c = \frac{S_{\text{np}}T_c I - \frac{1}{2}I^2(R+R_c) - K(T_h - T_c)}{S_{\text{ceramic}}} \quad (\text{S15})$$

where S_{ceramic} is the area of ceramic plate.

5. When discussing Q_c max in figure 4b, it is helpful to state in terms of W/cm^2 for easier comparison with modules having different sizes.

Response: We totally agree with the viewer that comparing the cooling power density (q_c) is a more appropriate approach for assessing the cooling capacity of different devices. We have modified the Fig. 4b and the relate discussion:

Fig. R6. The cooling power density (q_c) as a function of current for $\text{CGBi}_{1.5}/\text{Bi}_{0.5}\text{Sb}_{1.5}\text{Te}_3$ based modules, in comparison to the 7-pair SPS $\text{Mg}_3\text{Bi}_{1.5}\text{Sb}_{0.5}/\text{Bi}_{0.5}\text{Sb}_{1.5}\text{Te}_3$ cooling module with the normalized cross section.

6. It is claimed that the electrical resistivity is stable after 16 months, which is very stable. Is there any accelerated way for such testing? Do we have to wait for 16 months for testing such stability?

Response: Many thanks to the reviewer for raising this point. The monitoring period for the sample in this study spanned 16 months, from its initiation to the submission of

the manuscript. Our objective is to develop a packaging scheme for the devices that ensures a long-time service. In this study, the coarse-grained materials were stored in a glove box and the room temperature electrical transport property was periodically tested. The enhanced stability of the samples observed in the work suggests that the long-term service of the devices could be realized through appropriate packaging.

To accelerate the stability testing of thermoelectric devices, high temperature annealing can be employed. By subjecting the devices to couple of weeks of testing under constant working currents at elevated temperatures, such as 373 K, the thermal stability of both the $\text{Mg}_3(\text{Bi,Sb})_2$ -based materials and the contact interfaces of device can be evaluated. In order to protect them from chemical contamination, such as atmospheric moisture, during these tests, it becomes imperative to implement appropriate packaging for the device.

Minor points:

1. Is the $\text{ZrO}_2/\text{Ta}/\text{SiO}_2$ three-layer nested-crucible commercially available? If not, more description on this will be helpful for others who are interested to use similar strategy.

Response: Many thanks for your question. ZrO_2 crucible, Ta tube and quartz tube are three types of containers. All of them are commercially available. The photograph of the crucibles is included in supplementary materials of this version.

Fig. S1. The pictures of ZrO_2 crucible, sealed Ta tube containing ZrO_2 crucible, and sealed quartz crucible containing sealed Ta tube.

2. The yellow highlights throughout the manuscript maybe misunderstood, please remove them in the final version.

Response: Many thanks for your suggestions. Now those highlights have been removed. According to the comments from the reviewers, the new changes to the manuscript and supporting information are highlighted in this version.

3. The sentence in the introduction "... with grain size up to 1.0 mm, which approaches two thirds of a thermoelectric unit leg in the devices." Can be improved for clarity. I would suggest "... with grain size up to 1.0 mm. Such grain size encompasses almost two-third of the length of a thermoelectric leg in the devices"

Response: Thank you very much for your suggestions. We have revised our language accordingly.

4. "Then kept swinging for 2 h to allow the complete reaction of Mg/Bi/Sb and the formation of a uniform liquid phase." Can be: "The molten sample was swung continuously for 2 h in the rocking furnace to allow for the complete reaction of Mg/Bi/Sb and the formation of a uniform liquid phase."

Response: Thank you very much for your suggestions. We have revised our language accordingly.

5. The max dT during cooling is likely an underestimation, since it seems that the fill factor and aspect ratio have not yet been optimized in the 7-pair device.

Response: Many thanks to the reviewer for your reminding. According to the theoretical prediction, within a small range of optimal cross-sectional area ratio, the effect on the temperature difference is relatively small. When the sizes of the thermoelectric legs are at the optimum aspect ratio ($1.42 \times 1.42 \times 3 \text{ mm}^3$ for n-type CGBi_{1.5} legs and $1.25 \times 1.25 \times 3 \text{ mm}^3$ for p-type Bi_{0.5}Sb_{1.5}Te₃ legs), the predicted maximum temperature difference ΔT_{\max} is 71.1 K.

When the size of the n-type CGBi_{1.5} legs is slightly altered to $1.3 \times 1.3 \times 3 \text{ mm}^3$ for the construction of the cooling device, the ΔT_{\max} only decreases slightly to 70.8 K, a

difference of merely 0.3 K from the predicted value of 71.1 K. Although the device performance can be further improved by optimizing both the aspect ratio and design of the ceramic plates, the performance of current cooling device is already sufficient to reflect the properties of our materials.

6. “high reproductivity” should be “high reproducibility”

Response: Thank you for your correction. We have revised it accordingly.

7. Inset in Fig 1c, 3d are too small and low resolution.

Response: Thank you for your suggestions. The modified insets in Fig 1c and 3d are shown in the revised manuscript.

References:

1. Mao J, *et al.* High thermoelectric cooling performance of n-type Mg_3Bi_2 -based materials. *Science* **365**, 495-498 (2019).
2. Lei J, *et al.* Efficient lanthanide Gd doping promoting the thermoelectric performance of Mg_3Sb_2 -based materials. *Journal of Materials Chemistry A* **9**, 25944-25953 (2021).
3. Wang QQ, *et al.* In-Situ Loading Bridgman Growth of $\text{Mg}_3\text{Bi}_{1.49}\text{Sb}_{0.5}\text{Te}_{0.01}$ Bulk Crystals for Thermoelectric Applications. *Adv. Electron. Mater.* **8**, 2101125 (2022).
4. Wood M, Kuo JJ, Imasato K, Snyder GJ. Improvement of Low-Temperature zT in a Mg_3Sb_2 - Mg_3Bi_2 Solid Solution via Mg-Vapor Annealing. *Adv Mater* **31**, e1902337 (2019).
5. Ying P, *et al.* A robust thermoelectric module based on $\text{MgAgSb}/\text{Mg}_3(\text{Sb,Bi})_2$ with a conversion efficiency of 8.5% and a maximum cooling of 72 K. *Energy Environ. Sci.* **15**, 2557-2566 (2022).
6. Imasato K, Kang SD, Snyder GJ. Exceptional thermoelectric performance in $\text{Mg}_3\text{Sb}_{0.6}\text{Bi}_{1.4}$ for low-grade waste heat recovery. *Energy Environ. Sci.* **12**, 965-971 (2019).

REVIEWER COMMENTS

Reviewer #1 (Remarks to the Author):

Thanks to the detailed explanation and some additional data shown in both SI and main text. Their clarification of the details on the experimental procedure, data analysis and the validation of their claim should be helpful to understand the results of this work. The added explanations are also important to improve accuracy and their statements are strengthened with the additional data. As the revised paper is polished and more understandable, I would recommend accepting this article to Nature Communications.

Reviewer #2 (Remarks to the Author):

The authors have given a clear response. However, there is still some concerns should be addressed before acceptance.

1. This manuscript demonstrates that the Mg-Sb based TE module has an equal level performance of commercial Bi₂Te₃-based TE modules. To make the article more convincing, the data of maximum cooling temperature differences with a variety of hot-side temperatures for the commercial Bi₂Te₃-based modules from the leading enterprise of KELK and Ferrotec should be involved in Fig.4a, which is the flagship of the thermoelectric industry.

2. The Fig.4f presents the aging time dependence of electrical resistivity of CGBi_{1.5} materials. In the response letter, the authors claim that the coarse-grained bars (~ 1.7×2.7×10 mm³) stored in a glove box and tested the room-temperature electrical transport property every week. And the measurements were performed under a helium atmosphere. These storage and testing conditions are quite different from that of the TE modules. Furthermore, the TE module consists of p- and n-type TE materials, electrodes, and ceramic substrates, which structure is more complex than that of the coarse-grained bars. So, the Fig.4f cannot represent the stability of Mg-Sb TE module. And it should not be put in Fig.4 because the caption of Fig. 4 is "Cooling and power generation performances of Mg₃Bi_{2-x}Sb_x/Bi_{0.5}Sb_{1.5}Te₃ (x=0.5,0.75) based TE modules." Here, the testing data of thermal duration and/or thermal cycling experiments of TE module should be provided. Otherwise, the Fig. 4f should be taken out of Fig. 4 to avoid misunderstanding.

Reviewer #3 (Remarks to the Author):

I have had the opportunity to carefully read the authors' reply. I commend the efforts put in by the authors to improve their manuscript. Figure R5 is helpful and the explanation adds value to this work. Additional details such as Table S4 and Calculation of cooling performance of thermoelectric devices in the supporting information are helpful.

I am therefore happy to recommend publication of the current version.

Response to Reviewer Reports

All changes to the manuscript and supporting information are indicated in blue text below and highlighted in the respective documents.

Reviewer #1:

Thanks to the detailed explanation and some additional data shown in both SI and main text. Their clarification of the details on the experimental procedure, data analysis and the validation of their claim should be helpful to understand the results of this work. The added explanations are also important to improve accuracy and their statements are strengthened with the additional data. As the revised paper is polished and more understandable, I would recommend accepting this article to Nature Communications.

Response: We are very grateful to the reviewer for her/his generous words and for the thoughtful and thorough review of our work, which improved the way how the study is presented significantly.

Reviewer #3:

I have had the opportunity to carefully read the authors' reply. I commend the efforts put in by the authors to improve their manuscript. Figure R5 is helpful and the explanation adds value to this work. Additional details such as Table S4 and Calculation of cooling performance of thermoelectric devices in the supporting information are helpful.

I am therefore happy to recommend publication of the current version.

Response: We are very appreciative of the reviewer for her/his supportive statement. We thank the reviewer for all the comments and favorable suggestions, which improved the quality of our manuscript solidly.

Reviewer #2:

The authors have given a clear response. However, there is still some concerns should be addressed before acceptance.

Response: We are very grateful to the reviewer for the positive and supportive perspective of our previous responses and revisions, and we consider the comments in detail as below.

1. This manuscript demonstrates that the Mg-Sb based TE module has an equal level performance of commercial Bi₂Te₃-based TE modules. To make the article more convincing, the data of maximum cooling temperature differences with a variety of hot-side temperatures for the commercial Bi₂Te₃-based modules from the leading enterprise of KELK and Ferrotec should be involved in Fig.4a, which is the flagship of the thermoelectric industry.

Response: We thank the reviewer for raising this point on further explicit comparisons, and we completely agree that including the suggested data are helpful and important in appreciating the significance of our work and others. Thus, we visited the websites of KELK and Ferrotec, and searched for the cooling temperature difference of their cooling devices with similar sizes to those in our work ($\sim 10 \times 10 \times 5.6 \text{ mm}^3$) (as indicated in Fig. R2 and Fig. R3). Accordingly, in the revised manuscript, the current state-of-art parameters have been included in Fig. 4a with corresponding discussions being added as below:

Fig. R1. The comparison of maximum cooling temperature differences between Mg₃Bi_{2-x}Sb_x/Bi_{0.5}Sb_{1.5}Te₃ ($x=0.5, 0.75$) TE modules and commercial Bi₂Te₃-based module with a variety of hot-side temperatures T_h .

The difference in ΔT_{\max} curve between the modules reported in this work and commercial Bi_2Te_3 -based devices can be attributed to the higher near-room-temperature thermoelectric properties of Bi_2Te_3 and the advanced contact layer fabrication technique in the leading thermoelectric enterprise.

■ Multi-purpose modules

Type	Model No.	I _{max} (A)	V _{max} (volts)			ΔT _{max} (°C)			Q _{cmax} (watts)			Top ceramics		Bottom ceramics		Height H(mm)	Height tolerance (±mm)	Ceramic material	Metallization		Lead Wire
			Th=27°C	Th=50°C	Th=70°C	Th=27°C	Th=50°C	Th=70°C	Th=27°C	Th=50°C	Th=70°C	W (mm)	L1(mm)	W (mm)	L2(mm)				Nil	Cu-Ni-Au	
Multi-purpose modules	KSM-04007E	4.0	0.8	0.9	1.0	68.0	77.0	84.0	1.9	2.1	2.3	10.3	10.3	10.3	12.2	4.7	0.2	Alumina(Al ₂ O ₃)	○		20AWG
	KSM-04017E	4.0	2.0	2.2	2.5	68.0	77.0	84.0	4.7	5.1	5.5	15.0	15.0	15.0	17.0	4.7	0.2	Alumina(Al ₂ O ₃)	○		20AWG
	KSM-04031E	4.0	3.6	4.1	4.5	68.0	77.0	84.0	8.6	9.3	9.9	20.0	20.0	20.0	22.0	4.7	0.2	Alumina(Al ₂ O ₃)	○		20AWG
	KSM-04071E	4.0	8.3	9.3	10.3	68.0	77.0	84.0	19.6	21.4	22.7	30.0	30.0	30.0	31.5	4.7	0.2	Alumina(Al ₂ O ₃)	○		20AWG
	KSM-04127E	4.0	14.8	16.7	18.4	68.0	77.0	84.0	35.0	38.2	40.6	40.0	40.0	40.0	41.5	4.7	0.2	Alumina(Al ₂ O ₃)	○		20AWG
	KSM-06007E	6.0	0.8	0.9	1.0	67.0	76.0	83.0	2.9	3.2	3.4	10.3	10.3	10.3	12.2	3.8	0.2	Alumina(Al ₂ O ₃)	○		20AWG
	KSM-06017E	6.0	2.0	2.2	2.5	67.0	76.0	83.0	7.0	7.6	8.1	15.0	15.0	15.0	17.0	3.8	0.2	Alumina(Al ₂ O ₃)	○		20AWG
	KSM-06031E	6.0	3.6	4.1	4.5	67.0	76.0	83.0	12.7	13.9	14.8	20.0	20.0	20.0	22.0	3.8	0.2	Alumina(Al ₂ O ₃)	○		20AWG
	KSM-06071E	6.0	8.2	9.3	10.2	67.0	76.0	83.0	29.0	31.7	33.7	30.0	30.0	30.0	31.5	3.8	0.2	Alumina(Al ₂ O ₃)	○		20AWG
	KSM-06127E	6.0	14.7	16.6	18.3	67.0	76.0	83.0	51.9	56.7	60.3	40.0	40.0	40.0	41.5	3.8	0.2	Alumina(Al ₂ O ₃)	○		20AWG

Fig. R2. Screenshot of cooling performance of thermoelectric devices from the website of KELK (<https://www.kelk.co.jp/english/products/thermo.html>).

(Th=50°C)

TEM Model No.	I _{max} (A)	V _{max} (V)	ΔT _{max} (°C)	Q _{cmax} (W)	SIZE(mm)			
					W	L1	L2	H
20013/017/030B	3.0	2.4	83	3.8	11.5	11.5		2.90
20013/023/030B	3.0	3.3	83	5.1	7.4	22.4		2.90
20005/017/040B	4.0	2.4	83	5.1	15.1	15.1		3.10
20013/031/040B	4.0	4.4	83	9.2	15.1	15.1		2.90
20005/035/040B	4.0	5.0	83	10	15.1	29.8		3.95
20015/063/040B	4.0	9.0	83	19	20.1	39.7		3.95
20013/071/040B	4.0	10.1	83	21	22.4	22.4		2.90
20003/031/085B	8.5	4.4	83	20	20.0	20.0		3.75

Fig. R3. Screenshot of cooling performance of thermoelectric devices from the website of Ferrotec (<http://www.ferrotec.com.cn/products/productinfo/84.html>).

2. The Fig.4f presents the aging time dependence of electrical resistivity of $\text{CGBi}_{1.5}$ materials. In the response letter, the authors claim that the coarse-grained bars ($\sim 1.7 \times 2.7 \times 10 \text{ mm}^3$) stored in a glove box and tested the room-temperature electrical transport property every week. And the measurements were performed under a helium atmosphere. These storage and testing conditions are quite different from that of the TE modules. Furthermore, the TE module consists of p- and n-type TE materials, electrodes, and ceramic substrates, which structure is more complex than that of the coarse-grained bars. So, the Fig.4f cannot represent the stability of Mg-Sb TE module. And it should not be put in Fig.4 because the caption of Fig. 4 is “Cooling and power generation performances of $\text{Mg}_3\text{Bi}_{2-x}\text{Sb}_x/\text{Bi}_{0.5}\text{Sb}_{1.5}\text{Te}_3$ ($x=0.5,0.75$) based TE modules.” Here, the

testing data of thermal duration and/or thermal cycling experiments of TE module should be proved. Otherwise, the Fig. 4f should be taken out of Fig. 4 to avoid misunderstanding.

Response: We are highly appreciative of the reviewer for her/his careful and rigorous scrutiny of the content and data of Fig. 4f. We totally agree that we should provide more details about the stability of our thermoelectric devices. Accordingly, we have included the device stability data in the updated Fig. 4f, and added the corresponding discussion in the revised manuscript as follows:

Fig. R4. Aging time dependent maximum cooling temperature differences of CGBi_{1.5}/Bi_{0.5}Sb_{1.5}Te₃ devices and electrical resistivity of CGBi_{1.5} samples.

As shown in Fig. 4f, the maximum temperature differences of CGBi_{1.5}/Bi_{0.5}Sb_{1.5}Te₃ devices only reduced by *ca.* 1 K after being stored in the glove box for nearly 6 months. In addition, we also observed that with reduced Mg1 vacancies, the electrical resistivity of coarse-grained CGBi_{1.5} bulk materials stored in the glove box remained unchanged over 16 months, which, to the best of our knowledge, has been the best record to date. These reinforce excellent stabilities of both the thermoelectric materials and electrodes of Mg₃Bi_{2-x}Sb_x/Bi_{0.5}Sb_{1.5}Te₃ (*x*=0.5,0.75) modules. The results also provide feasibility to further improve device-level stabilities through subsequent device packaging strategies, underpinning potential real-world applications.

REVIEWERS' COMMENTS

Reviewer #2 (Remarks to the Author):

The added explanations improved the accuracy and strengthened the statements. I would recommend accepting this article to Nature Communications.

Response to Reviewer Reports

Reviewer #2:

The added explanations improved the accuracy and strengthened the statements. I would recommend accepting this article to Nature Communications.

Response: We are very grateful to the reviewer for her/his favorable suggestions, which greatly improved the quality of our work.